# From Counterfactuals to Trees:
# Competitive Analysis of Model Extraction Attacks

**Awa Khouna**
Polytechnique Montréal
awa.khouna@polymtl.ca

**Julien Ferry**
Polytechnique Montréal
julien.ferry@polymtl.ca

**Thibaut Vidal**
Polytechnique Montréal
thibaut.vidal@polymtl.ca

## Abstract

The advent of Machine Learning as a Service (MLaaS) has heightened the trade-off between model explainability and security. In particular, explainability techniques, such as counterfactual explanations, inadvertently increase the risk of model extraction attacks, enabling unauthorized replication of proprietary models. In this paper, we formalize and characterize the risks and inherent complexity of model reconstruction, focusing on the "oracle" queries required for faithfully inferring the underlying prediction function. We present the first formal analysis of model extraction attacks through the lens of competitive analysis, establishing a foundational framework to evaluate their efficiency. Focusing on models based on additive decision trees (e.g., decision trees, gradient boosting, and random forests), we introduce novel reconstruction algorithms that achieve provably perfect fidelity while demonstrating strong anytime performance. Our framework provides theoretical bounds on the query complexity for extracting tree-based models, offering new insights into the security vulnerabilities of their deployment.

## 1 Introduction

Recent research has shown that sharing trained machine learning (ML) models can lead to the reconstruction of sensitive training data, posing significant privacy risks [see, e.g., Boenisch et al., 2023, Carlini et al., 2024, Ferry et al., 2024]. Applications in fields such as medical diagnostics, financial services, and personalized advertising often handle large amounts of private data, making them attractive targets for data reconstruction attacks. These attacks exploit vulnerabilities in the model to recover confidential information from the training dataset, thereby undermining the privacy guarantees that organizations seek to uphold. Consequently, organizations may prefer to utilize Machine Learning as a Service (MLaaS) to leverage powerful models without directly exposing them, balancing the benefits of advanced analytics with the need to protect sensitive information.

While MLaaS platforms provide accessible and scalable ML solutions, they must address the growing demand for explainability in their decision-making processes. Regulatory frameworks such as the EU AI Act's Article 13[1] further mandate greater transparency across a wide range of applications. In response, MLaaS providers increasingly incorporate explainability techniques to elucidate model behavior and ensure fairness. Notably, counterfactual explanations specify the changes an input example must undergo to alter its prediction, thereby directly offering a form of recourse in many real-life applications. However, studies have shown that querying a model's explanations can enable attackers to replicate its parameters and architecture, effectively copying the original model [Tramèr et al., 2016, Wang et al., 2022, Aïvodji et al., 2020, Öksüz et al., 2024]. This reveals a critical tension between the need for transparency and the protection of model integrity and intellectual property.

---

[1]https://artificialintelligenceact.eu/article/13/

39th Conference on Neural Information Processing Systems (NeurIPS 2025).

Model extraction attacks were proposed against a variety of ML models in recent years [Oliynyk et al., 2023]. While very few of them are *functionally equivalent* (i.e., they provably reconstruct the black-box model's decision boundary), they often rely on strong assumptions, such as access to a leaf identifier in the case of decision tree models [Tramèr et al., 2016]. Moreover, the majority of the literature focuses solely on empirically evaluating the fidelity of the extracted model w.r.t. the target black-box, lacking a rigorous framework for analyzing attack complexities and thoroughly characterizing their worst-case scenarios. Finally, while counterfactual explanations constitute a promising attack surface and were exploited to conduct model extraction attacks [Aïvodji et al., 2020, Wang et al., 2022, Dissanayake and Dutta, 2024], existing approaches rely on training surrogate models without functional equivalence guarantees.

In this study, we address these limitations through the following key contributions:

- We define a rigorous framework to characterize the complexity of model extraction attacks, utilizing competitive analysis (a notion from online optimization) to evaluate the difficulty of reconstructing models under various conditions.

- We introduce a novel algorithm (TRA) specifically designed to efficiently extract axis-parallel decision boundary models (including, but not limited to, tree ensemble models) through locally optimal counterfactual explanations.

- We provide a comprehensive theoretical analysis of our proposed method, offering guarantees on query complexity and demonstrating 100% fidelity in the extracted models.

- We conduct extensive experiments to validate our theoretical findings, presenting an average-case and anytime performance analysis of TRA compared to state of the art reconstruction methods. These experiments not only confirm our theoretical results, but also provide practical insights into the effectiveness and limitations of our approach.

These contributions collectively highlight and permit us to better characterize security vulnerabilities in deploying explainable tree ensembles.

## 2 Online Discovery, Model Extraction Attacks and Competitive Analysis

Online discovery problems have long been a focus of research in theoretical computer science, where the goal is to uncover the structure of an unknown environment through a sequence of queries or observations [Ghosh and Klein, 2010, Deng et al., 1991]. A classic example arises in map exploration: an agent (e.g., a robot) navigates a space cluttered with obstacles, with only a limited "line of sight" at each position. The agent's objective is to construct a complete representation (e.g., map) of its surroundings while minimizing resources such as travel distance or exploration time.

Model extraction attacks on MLaaS platforms exhibit striking parallels to these online exploration tasks. In a typical model extraction attack, an adversary queries a predictive model (the "black box") to gain information about its internal decision boundaries, effectively learning the decision function through a limited set of inputs and outputs. Drawing an analogy to the map exploration scenario, each query in a model extraction attack can be likened to a "probe" in the space of features that reveals partial information about the region—namely, the predicted label or a counterfactual explanation identifying the closest boundary capable of changing the prediction. Figure 1 illustrates this connection by contrasting a rover's sensor sweep in a polygon exploration task with a query to locate a counterfactual explanation in a machine-learning model.

**Online discovery problems & Model extraction attacks.** Online discovery typically assumes an agent that can move freely in the physical world while receiving feedback about obstacles in its vicinity. In model extraction, the "environment" is the model's input space, and the queries return a point that lies on the nearest decision boundary (or provides the counterfactual boundary itself). Thus, while map exploration may allow richer geometric observations (e.g., an entire sensor sweep of obstacles), counterfactual-based model extraction often yields more constrained information (e.g., only the nearest boundary for a given input). Despite these differences, both problems share a common hallmark: the true structure (environment or decision boundaries) is unknown *a priori* and must be inferred *online* via carefully chosen queries.

**Competitive Analysis.** A central tool for analyzing online discovery problems is *competitive analysis* [Karlin et al., 1986], which compares the performance of an *online* algorithm — one that

Robot exploration                       Model extraction

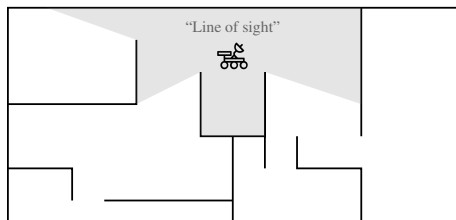 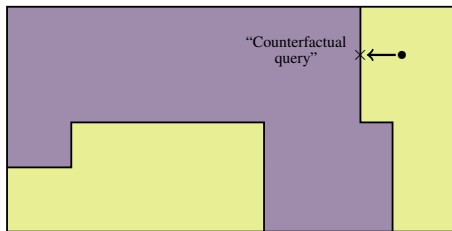

Figure 1: Illustration of the connection between online discovery problems and model extraction attacks. **Left** (adapted from Tee and Han [2021]): an autonomous robot maps an unknown 2D environment (e.g., a house) with limited-range sensors (e.g., LIDAR and laser distance measurements). **Right**: model extraction attacks recover the model's decision boundary via counterfactual queries.

adapts its decisions based solely on information acquired so far — to an optimal *offline* algorithm with complete foresight. Formally, we measure the ratio between: (i) the complexity (e.g., number of queries, computational cost) incurred by the online algorithm to achieve its goal and (ii) the minimal complexity that an offline algorithm, with complete foresight, would require to accomplish the same task. A constant ratio implies a *constant-competitive* algorithm; in more complex settings, the ratio may grow with problem parameters.

By applying competitive analysis to model extraction attacks, we can quantify how many queries (i.e., counterfactuals or label predictions) are needed to guarantee perfect fidelity in model reconstruction under worst-case conditions, complementing empirical investigations. Moreover, competitive analysis encourages us to ask: *How many queries, relative to an all-knowing attacker, does one need in order to prove with certainty that a specific model has been recovered?* This yields a principled measure of the difficulty of extracting tree-based models, analogous to classic results in online map discovery [Deng et al., 1991, Hoffmann et al., 2001, Ghosh and Klein, 2010, Fekete and Schmidt, 2010].

Overall, this perspective paves the way for a unified view: model extraction attacks can be seen as online exploration in the feature space. Our work, therefore, provides new theoretical results for tree-based model extraction, and invites cross-pollination between the literature on online discovery algorithms and emerging threats in machine learning security.

## 3 Method

### 3.1 Problem Statement

We consider an input as an $m$-dimensional vector in the input space $\mathcal{X} = \mathcal{X}_1 \times \mathcal{X}_2 \times \cdots \times \mathcal{X}_m \subseteq \mathbb{R}^m$, and the output belongs to the categorical space $\mathcal{Y}$. Let $\mathcal{F}$ denote the set of all axis-parallel decision boundary models, including decision trees and their ensembles, such as random forests [Breiman, 2001]. A machine learning (ML) classification model $f \in \mathcal{F}$ is defined as a function $f : \mathcal{X} \mapsto \mathcal{Y}$. For simplicity and without loss of generality, we focus our discussion on decision trees, as any axis-parallel decision boundary model can be represented as a decision tree [Vidal and Schiffer, 2020]. The input space $\mathcal{X}$ may comprise both categorical and continuous features.

**Definition 3.1.** Let $d$ be a distance function and $\mathcal{P}(\mathcal{X})$ denote the set of all subsets of $\mathcal{X}$. A counterfactual explanation oracle $\mathcal{O}_d$ is defined as a function $\mathcal{O}_d : \mathcal{F} \times \mathcal{X} \times \mathcal{P}(\mathcal{X}) \mapsto \mathcal{X}$. For a given model $f \in \mathcal{F}$, an instance $x \in \mathcal{X}$, and an input subspace $\mathcal{E} \subseteq \mathcal{X}$, $x' = \mathcal{O}_d(f, x, \mathcal{E})$ is a counterfactual explanation such that $x' \in \mathcal{E}$ and $f(x') \neq f(x)$. This counterfactual is *locally optimal* if:

$$\exists\, \epsilon > 0 \text{ such that } \forall v \in \mathbb{R}^m \setminus \{0\},\ \|v\| \leq \epsilon, \quad f(x) = f(x' + v) \quad \text{or} \quad d(x, x' + v) \geq d(x, x').$$

Intuitively, this condition ensures that any small perturbation ($v$) of $x'$ either gives an invalid counterfactual (having the same label as $x$) or increases the distance to the original input $x$.

For an adversary with black-box access to a target model $f$ (*i.e.,* through a prediction API), a *model extraction attack* aims to retrieve the exact model's parameters [Tramèr et al., 2016]. However, this goal is often too strict as many models might encode the same prediction function and thus

remain indistinguishable through counterfactual or prediction queries. Consequently, a more tractable objective, known as *functionally equivalent extraction* (Definition 3.2), focuses on reconstructing a model encoding the exact same function over the input space [Jagielski et al., 2020].

**Definition 3.2.** A functionally equivalent extraction attack aims to reconstruct a model $\hat{f} \in \mathcal{F}$ such that it is functionally identical to the target model $f \in \mathcal{F}$ across the entire input space $\mathcal{X}$. Formally, the attack seeks to find $\hat{f}$ satisfying Equation (1) using as few queries as possible.

$$\forall x \in \mathcal{X}, \quad \hat{f}(x) = f(x) \tag{1}$$

A common way to empirically evaluate such attacks is through the *fidelity* [Aïvodji et al., 2020] of the model reconstructed by the attacker, coined the *surrogate model*. It is defined as the proportion of examples (from a given dataset) for which the surrogate agrees with the target model. To theoretically bound the efficiency of a model extraction attack, we additionally rely on the notion of *c-competitiveness* from the online discovery literature, formalized in Definition 3.3.

**Definition 3.3.** Let $\mathcal{A}$ denote an online model extraction attack algorithm. Define $Q_{\mathcal{A}}^{f}$ as the number of queries required by $\mathcal{A}$ to extract the decision boundary of model $f$, and let $Q_{opt}^{f}$ represent the minimal (optimal) number of queries necessary to extract $f$ by an omniscient offline algorithm. The algorithm $\mathcal{A}$ is said to be *c-competitive* if, for any model $f \in \mathcal{F}$:

$$Q_{\mathcal{A}}^{f} \leq c \cdot Q_{opt}^{f}$$

In this work, we focus on functionally equivalent model extraction attacks of axis-parallel decision boundary models. We assume that for each query $x$ to the API, the attacker obtains (i) the label $f(x)$ of the query and (ii) a locally optimal counterfactual explanation $x'$.

### 3.2 Tree Reconstruction Attack algorithm

We now introduce our proposed **Tree Reconstruction Attack (TRA)**, detailed in Algorithm 1. TRA is a divide-and-conquer based algorithm that aims to reconstruct a decision tree $f_n$ with $n$ split levels by systematically exploring the input space $\mathcal{X}$, using only a black-box API returning predictions and locally optimal counterfactuals. A split level is defined as a particular value for a given feature that divides the input space into two subspaces. Note that multiple nodes within different branches of a decision tree can share the same split level.

**Algorithm Overview.** TRA operates by maintaining a query list $\mathcal{Q}$ that initially contains the entire input space $\mathcal{X}$. The algorithm iteratively processes each input subset $\mathcal{E} \subseteq \mathcal{X}$ from $\mathcal{Q}$, until $\mathcal{Q}$ is empty, ensuring that all decision boundaries of the target model $f$ are identified and replicated in the reconstructed tree. More precisely, at each iteration, TRA first retrieves the subset $\mathcal{E}$ on top of the priority queue $\mathcal{Q}$. It computes its geometric center $x$ using the *center* function (line 5). It then queries the oracle $\mathcal{O}_d$ with the target model $f$, input $x$, and subset $\mathcal{E}$ to obtain a counterfactual explanation $x' = \mathcal{O}_d(f, x, \mathcal{E})$ (line 8). The set of feature indices where $x'$ differs from $x$, i.e., $\{i \mid x_i' \neq x_i\}$ is consequently identified. For each differing feature $i$, TRA splits the input subset $\mathcal{E}$ into two subspaces based on the split value $x_i'$ (SPLIT function, detailed in Algorithm 2 in the Appendix B). The resulting subspaces are added to $\mathcal{Q}$ for further exploration (line 9). If no counterfactual explanation $x'$ exists within $\mathcal{E}$, TRA assigns the label $y = f(x)$ to $\mathcal{E}$, indicating that it corresponds to a leaf node in the reconstructed tree (line 11).

**Illustrative Example.** To illustrate TRA's operation, consider the axis-parallel decision boundary model illustrated on the right side of Figure 2b. Initially, TRA begins with the entire input space $\mathcal{X} = [0, 1]^2$. In the first iteration, the algorithm selects the center point $x^{(1)} = (0.5, 0.5)$ of $\mathcal{X}$ and queries the counterfactual explanation oracle $\mathcal{O}_{\|.\|_2}$, which returns a counterfactual $x'^{(1)} = (0.5, 0.4)$ that differs from $x^{(1)}$ in the second feature ($x_2$). This results in the first split of the input space (to $\mathcal{E}_1 = [0, 1] \times ]0.4, 1]$ and $\mathcal{E}_2 = \mathcal{X} \setminus \mathcal{E}_1$) based on the condition $x_2 \leq 0.4$, as shown on the left side of Figure 2b. In the subsequent iterations, TRA focuses on the resulting subspaces. For example, within the subset where $x_2 > 0.4$, TRA identifies another split at $x_1 \leq 0.7$, further partitioning the space. After three iterations, the reconstructed decision tree (shown in Figure 2a) accurately captures part of the decision boundaries of the target model, effectively distinguishing between different regions in the input space. The gray hatched zones (or "?" nodes) represent regions that have not yet been explored and remain in the query list $\mathcal{Q}$.

**Algorithm 1** Tree Reconstruction Attack (TRA)

1: **Input:** Oracle $\mathcal{O}_d$, target model $f : \mathcal{X} \mapsto \mathcal{Y}$.
2: $\mathcal{Q} \leftarrow \{\mathcal{X}\}$ {Initialize query list with the entire input space}
3: **repeat**
4:    $\mathcal{E} \leftarrow \mathcal{Q}.pop(0)$ {Retrieve the next input subset to investigate}
5:    $x \leftarrow center(\mathcal{E})$ {Compute the center point of $\mathcal{E}$}
6:    $y \leftarrow f(x)$ {Obtain the label of the center point}
7:    **if** $\mathcal{O}_d(f, x, \mathcal{E})$ exists **then**
8:       $x' \leftarrow \mathcal{O}_d(f, x, \mathcal{E})$ {Obtain counterfactual explanation}
9:       $\mathcal{Q} \leftarrow insert(\text{SPLIT}(\mathcal{E}, x, x'))$ {Split $\mathcal{E}$ and add the resulting subspaces to $\mathcal{Q}$ (Alg. 2)}
10:   **else**
11:      Assign label $y$ to the subset $\mathcal{E}$ {No counterfactual found; $\mathcal{E}$ is a leaf node}
12:   **end if**
13: **until** $\mathcal{Q}$ is empty

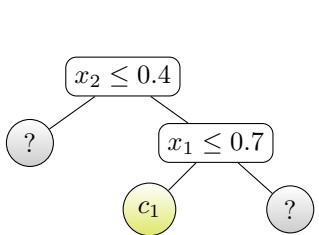 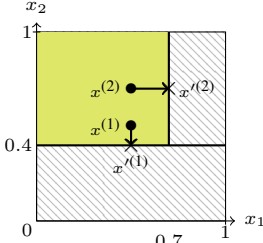 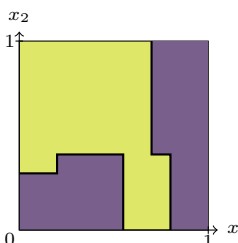

(a) Decision tree reconstructed by TRA after 3 iterations.

(b) Decision boundary of the model extracted by TRA after 3 iterations (Left) and of the target model $f : [0,1]^2 \to \{\, c_1 , c_2 \,\}$ (Right).

Figure 2: Illustrative example of the execution of TRA.

**Proposition 3.4.** *Let $f_n$ be a decision tree with $n$ split levels across a $m$-dimensional input space $\mathcal{X} = \mathcal{X}_1 \times \mathcal{X}_2 \times \cdots \times \mathcal{X}_m$. Denote $s_i$ as the number of split levels in $f_n$ over the $i$-th feature, such that $\sum_{i=1}^m s_i = n$. The worst-case complexity of Algorithm 1 is $O\left(\prod_{i=1}^m (s_i + 1)\right)$.*

**Corollary 3.5.** *The worst-case complexity of Algorithm 1 is $O\left(\left(1 + \frac{n}{m}\right)^m\right)$.*

The proofs of Proposition 3.4 and Corollary 3.5 are provided in the Appendix A. Proposition 3.4 establishes a first simple upper bound on the complexity of Algorithm 1. Intuitively, consider a two-dimensional decision boundary that partitions the space in a chessboard-like pattern of size $s_1 \times s_2$. A comprehensive mapping of such space necessitates at least $s_1 \times s_2$ queries (one in each sub-square), a requirement that holds for multi-class classification scenarios and in high-dimension.

**Query Selection Analysis.** An important hyperparameter of TRA is the strategy used to select query points within the input space. By default, TRA selects the geometrical center of the current input subset $\mathcal{E}$ as the query point. Alternatively, one could choose other points such as the lower/upper left/right corners, or even a random point. In the following, we present theoretical results analyzing the impact of different query selection strategies on the algorithm's performance. To this end, we leverage the notion of competitive analysis for online discovery problems discussed in Section 2.

**Proposition 3.6.** *For $(n, m) \in \mathbb{N}^2$, Algorithm 1 achieves a competitive ratio of $C_{TRA}^{(n,m)}$, defined as:*

$$C_{TRA}^{(n,m)} = \frac{2 \prod_{j=1}^m (s_j + 1) - 1}{n + 1} \leq \frac{2\left(1 + \frac{n}{m}\right)^m - 1}{n + 1},$$

*where $s_i$ is the number of split levels along the $i$-th feature within the tree $f_n$.*

**Proposition 3.7.** *For all $n > 0$ and $m \geq 2$, no divide-and-conquer-based algorithm can achieve a competitive ratio better than $C_{TRA}^{(n,m)}$.*

The proofs of Propositions 3.6 and 3.7 are provided in Appendix A. Proposition 3.6 provides the competitive ratio achieved by the TRA algorithm, while Proposition 3.7 establishes that the choice of

query position does not affect the competitive ratio for any divide-and-conquer algorithm iteratively partitioning the input space. These propositions demonstrate that TRA not only offers a competitive approach to tree reconstruction under various query selection strategies, but also sets a theoretical limit that cannot be surpassed by other methods with a similar divide-and-conquer structure.

**Anytime Behavior.** Since the query budget is often not known in advance and may vary depending on the target model, it is crucial for an extraction attack to operate in an *anytime* fashion—that is, to produce a usable classifier even if interrupted before completion. TRA satisfies this property by assigning provisional labels at each input space split (line 9 of Algorithm 1): one subregion inherits the query's label, the other the counterfactual's. This guarantees that a valid decision tree classifier is available at any point during execution. The quality of intermediate classifiers depends on the ordering of the priority queue $\mathcal{Q}$. In practice, preliminary experiments suggested good anytime performance using breadth-first search (BFS), which distributes the exploration evenly. While the design of alternative priority strategies is a promising research direction to enhance TRA's anytime performance, the choice of exploration order does not impact the total number of queries required for exact reconstruction —this is solely determined by the algorithm's divide-and-conquer structure. Finally, we note that the fraction of total volume corresponding to the leaf regions that have already been fully explored by TRA at a given iteration directly lower bounds the proportion of feature space for which functional equivalence can be guaranteed in an anytime manner. In the case of a uniform data distribution over the feature space, this value also lower bounds the anytime surrogate fidelity, and one could use it to early stop TRA as soon as a target fidelity level is achieved.

## 4    Experiments

We now empirically evaluate the efficiency and effectiveness of our proposed TRA extraction attack and benchmark it against existing model extraction techniques. We first introduce the experimental setup, before discussing the results.

### 4.1    Experimental Setup

**Datasets.** We use five binary classification datasets, selected from related works on model extraction attacks [Aïvodji et al., 2020, Wang et al., 2022, Tramèr et al., 2016] and encompassing a variety of feature types, dimensionalities, and classification tasks, as summarized in Table 1. More precisely, we consider the COMPAS dataset [Angwin et al., 2016], as well as the Adult Income (Adult), Default of Credit Card Clients (Credit Card), German Credit and Student Performance (SPerformance) datasets from the UCI repository [Dua and Graff, 2017]. Categorical features are one-hot encoded, while numerical, discrete (ordinal) and binary ones are natively handled by both tree building procedures and reconstruction attacks. Each dataset is partitioned into training, validation, and test sets with proportions of 60%, 20%, and 20%, respectively.

Table 1: Summary of the datasets used in our experiments. For each dataset, $m$ is the number of features after pre-processing, encompassing $m_N$ numerical, $m_B$ binary, $m_C$ categorical (before one-hot encoding) and $m_D$ discrete (ordinal) ones. Each of the $m_C$ categorical features is one-hot encoded into $c_j$ binary dimensions, where $c_j$ is the number of categories of feature $j$. As a result, the total number of features becomes: $m = m_N + m_B + m_D + \sum_{j=1}^{m_C} c_j$.

| Dataset | #Samples | $m$ | $m_N$ | $m_B$ | $m_C$ | $m_D$ |
|---|---|---|---|---|---|---|
| Adult | 45222 | 41 | 2 | 2 | 4 | 3 |
| COMPAS | 5278 | 5 | 0 | 3 | 0 | 2 |
| Credit Card | 29623 | 14 | 0 | 3 | 0 | 11 |
| German Credit | 1000 | 19 | 1 | 0 | 3 | 5 |
| SPerformance | 395 | 43 | 0 | 13 | 4 | 13 |

**Training the target tree-based models.** We train two types of tree-based target models implemented in the scikit-learn library [Pedregosa et al., 2011]: decision trees and random forests. For decision trees, we experiment with varying `max_depth` parameters ranging from 4 to 10, as well as trees without maximum depth constraint (`max_depth` set to `None`). The random forests experiments focus on the COMPAS dataset, employing different numbers of trees (5, 25, 50, 75 and 100) to assess

scalability and robustness. To prevent overfitting, we utilize the validation set for hyperparameter tuning and apply cost-complexity pruning where applicable. All the details of training procedures and hyperparameters configurations are discussed in Appendix C.1.

**Baselines.** We benchmark TRA against three state-of-the-art model extraction attacks. First, PATHFINDING [Tramèr et al., 2016] is the only functionally equivalent model extraction attack against decision trees. While it does not rely on counterfactual examples, it assumes access to a leaf identifier indicating in which leaf of the target decision tree the query example falls. It is thus not applicable to random forests. Second, CF [Aïvodji et al., 2020] leverages counterfactual explanations to build a labeled attack set and train a surrogate model mimicking the target one. Third, DUALCF [Wang et al., 2022] enhances the CF approach by additionally computing the counterfactuals of the counterfactuals themselves, which has been shown to improve fidelity. We adapt the number of queries (which must be pre-fixed for both CF and DUALCF) to the complexity of the target model as it is set to 50 times the number of nodes in the target decision tree. PATHFINDING is configured with $\epsilon = 10^{-5}$ (pre-fixed precision of the retrieved splits) to achieve approximate functional equivalence.

We evaluate three surrogate model variants for CF and DUALCF: a multilayer perceptron (MLP), and two models from the same hypothesis class as the target model (i.e., a decision tree or a random forest), one of them sharing the exact same hyperparameters, and the other using default hyperparameter values. These variants reflect different levels of adversarial knowledge: the hypothesis class, the exact hyperparameters, or neither. Both CF and DUALCF were originally evaluated using heuristic counterfactual explanations from the DiCE [Mothilal et al., 2020] algorithm. To assess the impact of explanation optimality on the attack's performance, and to ensure fair comparisons, we tested these baselines using either DiCE or the OCEAN framework [Parmentier and Vidal, 2021], which formulates counterfactual search as a mixed-integer linear program and guarantees optimality.

Complete experimental results across all configurations of CF and DUALCF are reported in Appendix C.2. They demonstrate that fitting surrogate models of the same hypothesis class facilitates the extraction of both decision trees and random forests. However, knowledge of their hyperparameters does not provide any advantage to either attack. Finally, the non-optimal counterfactuals provided by DiCE lead to better fidelity results than the optimal ones computed by OCEAN for these two attacks. This can be explained by their heuristic nature, which leads to the building of more diverse counterfactuals, not necessarily lying next to a decision boundary. In the next section, we display results only for the best-performing configuration for both CF and DUALCF, achieved by training a surrogate model of the same hypothesis class and using DiCE counterfactuals.

**Evaluation.** We assess each model extraction attack using two metrics: *fidelity* and *number of queries* made to the prediction and counterfactual oracle API during the attack. Fidelity measures the proportion of inputs for which the extracted model agrees with the target model, quantifying attack success. We compute fidelity over 3000 points sampled uniformly from the input space, providing a broad evaluation across the entire feature domain. For completeness, we also report fidelity measured on a test set (i.e., drawn from the data distribution) for our experiments using random forest target models, in Appendix C.3, which shows the same performance trends.

**Counterfactual oracle.** As discussed in Section 3.2, TRA requires the use of a locally optimal counterfactual oracle. Since global optimality is a *sufficient* condition, we use the popular OCEAN oracle for simplicity in our main experiments with TRA. In Appendix D, we report the performance of TRA using a simple heuristic oracle that produces locally optimal explanations. These experiments show that the choice of oracle (optimal or heuristic) has minimal impact on reconstruction performance.

All experiments are run on a computing cluster with homogeneous nodes using Intel Platinum 8260 Cascade Lake @ 2.4GHz CPU. Each run uses four threads and up to 4GB of RAM each (multi-threading is only used by the OCEAN oracle). We repeat each experiment five times with different random seeds and report average values. The source code to reproduce all our experiments and figures is accessible at `https://github.com/vidalt/Tree-Extractor`, under an MIT license.

## 4.2 Results

**Result 1. TRA outperforms existing approaches in terms of number of queries and anytime fidelity to extract decision trees.** Figure 3 presents the average fidelity of surrogate models obtained from the four studied model extraction attacks on decision trees, as a function of the number of queries, for the Adult and COMPAS datasets. Here, each point on a curve is the mean fidelity over

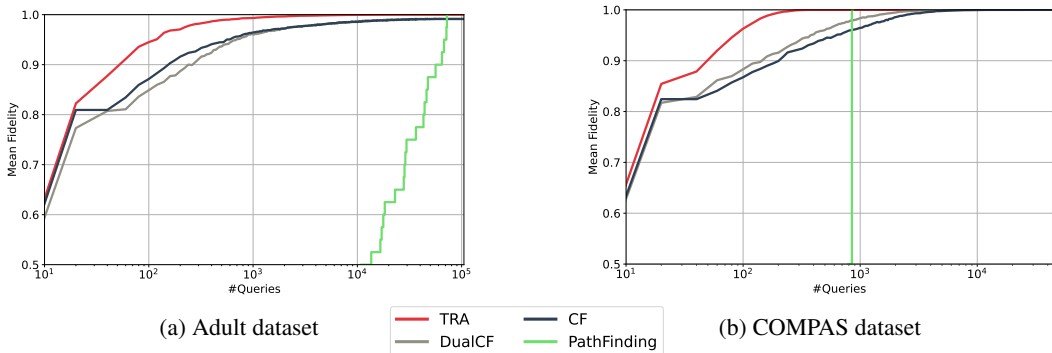

(a) Adult dataset       (b) COMPAS dataset

Figure 3: Anytime performance of all the considered model extraction attacks against decision trees.

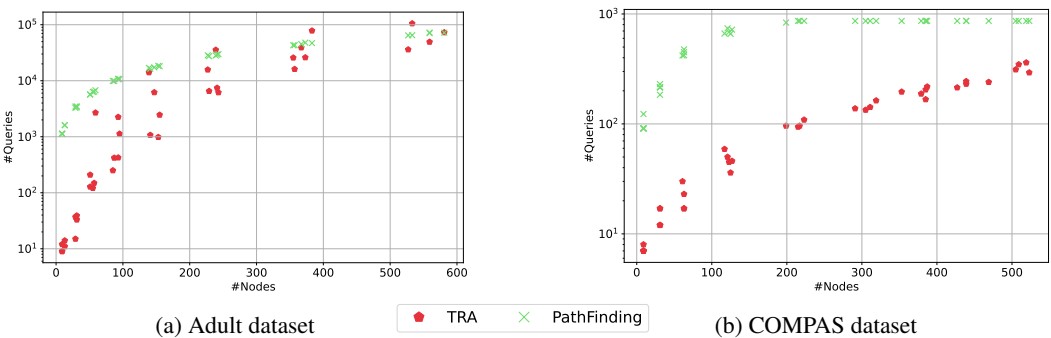

(a) Adult dataset       (b) COMPAS dataset

Figure 4: Performance of the functionally equivalent model extraction attacks against decision trees. We report the number of queries required to fully reconstruct the trees as a function of their size.

all 40 extraction tasks (eight tree depths × five seeds), so when it reaches 1.00, all target trees were perfectly reconstructed. Results for additional datasets, provided in Figure 12 (Appendix C.4), exhibit the same trends. Across all cases, TRA consistently achieves higher fidelity for any fixed query budget and converges to perfect fidelity orders of magnitude faster. Notably, unlike CF and DUALCF, TRA and PATHFINDING provide formal guarantees of functional equivalence.

**Result 2. TRA achieves state-of-the-art performance for functionally equivalent extraction of decision trees.** Figure 4 shows the number of queries required by PATHFINDING and TRA to achieve functionally equivalent model extraction, as a function of the number of nodes in the target models. Results for the Adult and COMPAS datasets are presented, with additional datasets provided in Figure 13 (Appendix C.4). TRA consistently requires orders of magnitude fewer queries than PATHFINDING to reconstruct the target models with perfect fidelity.

**Result 3. TRA theoretically and empirically out-performs existing approaches to extract random forests.** Figure 5 presents the average fidelity of the three considered model extraction attacks against random forests, plotted against the number of performed queries for the COMPAS dataset. The results show that TRA achieves higher fidelity with fewer counterfactual queries and converges significantly faster to perfect fidelity. Moreover, TRA is the only attack that certifies functional equivalence for tree ensembles. Additional results in Figure 14 (Appendix C.4) indicate that TRA scales efficiently with the size of the target random forest, as the number of required queries grows sub-linearly with the total number of nodes. This is due to structural redundancies in large forests, allowing the extracted model to be repre-

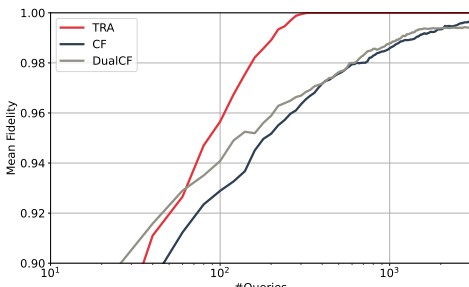

Figure 5: Anytime performance of the considered model extraction attacks against random forests, on the COMPAS dataset.

sented with perfect fidelity as a more compact decision tree [Vidal and Schiffer, 2020].

**Result 4. TRA is effective with either globally or locally optimal counterfactuals.** We report in Appendix D results from experiments where TRA is run with a simple oracle generating heuristic counterfactuals that are only locally optimal. The results (also provided within Figure 6 for the COMPAS dataset) show that TRA's performance is largely unaffected by the lack of global optimality. In some cases, locally optimal counterfactuals even improve early-stage (anytime) performance by introducing greater diversity in the explored input space. This highlights the practical applicability of TRA in real-world scenarios. Indeed, any valid but possibly sub-optimal counterfactual explanation can be post-processed (through bisection line-searches over each feature using simple prediction queries) into a locally optimal one lying on the decision boundary. Therefore, as long as the counterfactual oracle is reliable (i.e., returns a counterfactual whenever one exists), the returned explanation reveals a direction in which the prediction changes, which is then sufficient to locate a nearby decision boundary and conduct an efficient model extraction attack with TRA.

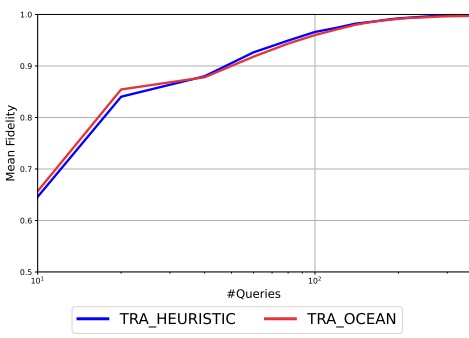 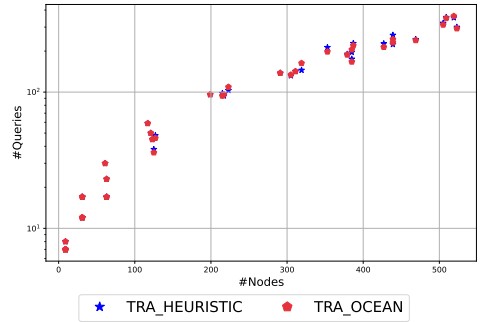

(a) Anytime performance of TRA: average surrogate fidelity as a function of the number of performed queries.

(b) Performance of TRA to achieve functional equivalence: each point represents the number of queries needed to fully reconstruct the trees.

Figure 6: Comparison of the performance of TRA using either OCEAN or a simpler heuristic counterfactual oracle (Algorithm 3 in Appendix D) to extract decision trees (COMPAS dataset).

**Result 5. TRA often requires far fewer queries in practice than its theoretical worst-case bound.** Across all datasets we evaluated, the number of counterfactual queries TRA uses to recover a functionally equivalent model is substantially below the worst case of Proposition 3.4. Table 2 reports the empirical query counts and the corresponding theoretical worst-case numbers for depth-9 decision trees (means over multiple random seeds). As evidenced, the former are consistently orders of magnitude smaller than the latter, suggesting that TRA is, on average, considerably more efficient than its theoretical worst-case.

Table 2: Empirical counterfactual query counts used by TRA versus the theoretical worst-case bound from Proposition 3.4 on decision trees of maximum depth 9. Empirical values are averaged over multiple random seeds.

| Dataset | Empirical # queries | Worst-case # queries (Prop. 3.4) |
|---|---|---|
| SPerformance | $1.16e + 03$ | $3.02e + 08$ |
| Adult | $3.70e + 04$ | $2.45e + 14$ |
| German Credit | $5.18e + 01$ | $2.86e + 03$ |
| Credit Card | $6.97e + 04$ | $6.49e + 11$ |
| COMPAS | $1.53e + 02$ | $1.07e + 03$ |

# 5   Related Works

The flourishing literature on privacy in machine learning encompasses a wide variety of inference attacks, considering different setups and objectives [Cristofaro, 2020, Rigaki and García, 2024]. This paper focuses on model extraction attacks [Tramèr et al., 2016], which aim at reconstructing the decision boundary of a black-box target model as accurately as possible, given a prediction API. As highlighted in recent surveys [Gong et al., 2020, Oliynyk et al., 2023], various attacks have been

proposed in recent years, targeting a broad spectrum of hypothesis classes. Hereafter, we focus on those targeting axis-parallel decision boundary models or exploiting counterfactual explanations.

Tramèr et al. [2016] propose the only functionally equivalent model extraction attack targeting regression or decision trees: PATHFINDING. It assumes that each query reply contains a unique identifier for the associated leaf. In a nutshell, PATHFINDING identifies the decision boundaries of each leaf in the target tree by varying the values of each feature. While effective, this method requires a large number of queries, though partial input queries can sometimes mitigate this overhead. In contrast, TRA does not make strong assumptions regarding the target model's prediction API, is able to extract any axis-parallel decision boundary model (beyond decision trees), and uses orders of magnitude fewer queries by exploiting counterfactual explanations.

While many recent works have focused on generating counterfactual explanations [Guidotti, 2024], these explanation techniques have also been shown to facilitate privacy attacks [Pawelczyk et al., 2023]. Aïvodji et al. [2020] introduce CF, a model extraction attack that leverages counterfactual explanations. Their approach constructs a labeled dataset by querying both predictions and counterfactuals from the target model, which is then used to train a surrogate. Wang et al. [2022] extend this method with DUALCF, which improves fidelity by additionally querying the counterfactuals of the counterfactual explanations. However, neither CF nor DUALCF provide fidelity guarantees, and they also do not leverage the structural properties of the target model. Dissanayake and Dutta [2024] employ polytope theory to show that a sufficient number of optimal counterfactual explanations can approximate convex decision boundaries. They propose a model extraction attack against locally Lipschitz continuous models, with fidelity guarantees dependent on the Lipschitz constants of the target and surrogate models. However, functional equivalence cannot be strictly certified, and as the authors acknowledge, these guarantees do not apply to axis-parallel models (such as decision trees), which lack local Lipschitz continuity and convexity. Also note that their approach relies on globally optimal counterfactuals (whereas TRA accommodates locally optimal ones).

Finally, while beyond the scope of this paper, other explanation-based model extraction attacks have been explored, including those relying on gradient-based [Milli et al., 2019, Miura et al., 2024] and other feature-based methods [Öksüz et al., 2024].

## 6 Conclusions and Discussion

We introduced the first functionally equivalent model extraction attack against decision trees and tree ensembles, leveraging locally optimal counterfactual explanations. In addition to its rigorous functional equivalence guarantee, the proposed method achieves higher fidelity than prior approaches while requiring fewer queries. We also leveraged well-established tools from online discovery to enable a formal analysis of model extraction, drawing an analogy between the two fields. We illustrated the applicability and relevance of this analysis by providing bounds on our attack's efficiency compared to the best achievable strategy, relying on the notion of competitive ratio. This perspective is essential for formally characterizing and comparing model extraction attacks.

Our study demonstrates that optimal counterfactual explanations can be systematically exploited to reconstruct tree ensembles via query APIs, as they inherently reveal decision boundaries. This raises significant concerns, especially as explainability is increasingly mandated by regulations. In many real-world applications, counterfactual explanations serve as a natural mechanism to meet transparency requirements by providing recourse information. We discuss the broader societal impacts of our work in Appendix E.

The research perspectives connected to our work are numerous. First, we believe that competitive analysis provides a valuable foundation for studying model extraction attacks, and future work should adopt the same lenses to evaluate other target models. Besides this, both the algorithms and their theoretical bounds could be refined. Improving input space exploration while mitigating worst-case query complexity is a key direction, including strategies such as dynamically reordering TRA's priority queue to avoid worst-case scenarios, or adopting completely different exploration methods (beyond divide-and-conquer-based algorithms). Finally, investigating the impacts of privacy-preserving mechanisms for counterfactual explanations [Vo et al., 2023] on model extraction attacks' success is a crucial direction towards conciliating trustworthiness and privacy through ML explainability APIs.

## Acknowledgments

This research was enabled by support provided by Calcul Québec and the Digital Research Alliance of Canada, as well as funding from the SCALE AI Chair in Data-Driven Supply Chains. It was also supported by the *Fonds de recherche du Québec – Nature et technologies (FRQNT)* through a Team Research Project *(327090)*.

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

# A   Proofs

**Proposition 3.4.** *Let $f_n$ be a decision tree with $n$ split levels across a $m$-dimensional input space $\mathcal{X} = \mathcal{X}_1 \times \mathcal{X}_2 \times \cdots \times \mathcal{X}_m$. Denote $s_i$ as the number of split levels in $f_n$ over the $i$-th feature, such that $\sum_{i=1}^{m} s_i = n$. The worst-case complexity of Algorithm 1 is $O\left(\prod_{i=1}^{m}(s_i + 1)\right)$.*

*Proof of Proposition 3.4.* We prove the proposition by induction on the number of split levels $n$. For clarity and precision, we denote the number of splits in the $i$-th dimension for a decision tree with $n$ split levels as $s_i^{(n)}$, rather than simply using $s_i$.

- **Base Case ($n = 1$):** For $n = 1$, there exists a single feature $j$ with $s_j^{(1)} = 1$ and $s_i^{(1)} = 0$ for all $i \neq j$. The number of queries required is at most $3 = 2s_j^{(1)} + 1 = O(s_j^{(1)})$.

- **Inductive Step:** Assume the statement holds for all trees with up to $n$ split levels. Consider a tree $f_{n+1}$ with $n + 1$ split levels. Let $1 \leq j \leq m$ be the feature index of the first detected split ($x_j \leq \alpha$) where $\alpha \in \mathbb{R}$, dividing $\mathcal{X}$ into two subspaces:

$$\mathcal{X}_j^1 = \{x \in \mathcal{X} \mid x_j \leq \alpha\}, \quad \mathcal{X}_j^2 = \{x \in \mathcal{X} \mid x_j > \alpha\}.$$

Each subspace contains subtrees $f_{n_1}$ and $f_{n_2}$ with $n_1, n_2 \leq n$ split levels, respectively. By the inductive hypothesis, the number of queries for each subtree is respectively $O\left(\prod_{i=1}^{m}(s_i^{(n_1)} + 1)\right)$ and $O\left(\prod_{i=1}^{m}(s_i^{(n_2)} + 1)\right)$.

Since $s_j^{(n+1)} = s_j^{(n_1)} + s_j^{(n_2)} + 1$ and $s_i^{(n_q)} = s_i^{(n+1)}$ for $i \neq j$ (and $q \in \{1, 2\}$), the total number of queries for $f_{n+1}$ is:

$$O\left(\prod_{i=1}^{m}(s_i^{(n+1)} + 1)\right).$$

Therefore, the query count for extracting $f_n$ is $O\left(\prod_{i=1}^{m}(s_i^{(n)} + 1)\right)$. $\qquad\qquad\square$

**Corollary 3.5.** *The worst-case complexity of Algorithm 1 is $O\left(\left(1 + \frac{n}{m}\right)^m\right)$.*

*Proof of Corollary 3.5.* We build on the worst-case complexity demonstrated in Proposition 3.4, and we solve the following optimization problem:

$$\max_{s_1^{(n)}, \ldots, s_m^{(n)}} \prod_{i=1}^{m}(s_i^{(n)} + 1) \quad \text{s.t.} \quad \sum_{i=1}^{m} s_i^{(n)} = n \quad \text{and} \quad s_i^{(n)} \geq 1 \; \forall i \in \{1, \ldots, m\}.$$

Since maximizing a positive value is equivalent to maximizing its logarithm, we transform the objective into:

$$\max_{s_1^{(n)}, \ldots, s_m^{(n)}} \sum_{i=1}^{m} \log(s_i^{(n)} + 1).$$

Applying the Karush-Kuhn-Tucker (KKT) conditions [Boyd and Vandenberghe, 2004], we find that the maximum occurs when $s_i^{(n)} = \frac{n}{m}$ for all $i$. Substituting back, the worst-case complexity becomes $O\left(\left(\frac{n}{m} + 1\right)^m\right)$. $\qquad\qquad\square$

**Proposition 3.6.** *For $(n, m) \in \mathbb{N}^2$, Algorithm 1 achieves a competitive ratio of $C_{TRA}^{(n,m)}$, defined as:*

$$C_{TRA}^{(n,m)} = \frac{2\prod_{j=1}^{m}(s_j + 1) - 1}{n + 1} \leq \frac{2\left(1 + \frac{n}{m}\right)^m - 1}{n + 1},$$

*where $s_i$ is the number of split levels along the $i$-th feature within the tree $f_n$.*

*Proof of Proposition 3.6 .* Let $n > 0$, $m \geq 1$, denote $\alpha_1, ..., \alpha_n$ the tree split levels (decision thresholds) and for each feature $j = 1, ..., m$, let $s_j$ represent the number of splits in the $j$-th dimension, ordered such that $s_1 \geq s_2 \geq \cdots \geq s_m$. Without loss of generality, assume that the split levels are grouped by dimension. Specifically, splits $\alpha_1$ to $\alpha_{s_1}$ occur in the first dimension, splits $\alpha_{s_1+1}$ to $\alpha_{s_1+s_2}$ in the second dimension, and so on. Additionally, within each dimension, the split levels are sorted in increasing order, i.e.,

$$\forall 1 \leq j \leq m, \quad \sum_{i=1}^{j-1} s_i + 1 \leq p \leq \sum_{i=1}^{j} s_i, \quad \alpha_p < \alpha_{p+1}.$$

1. *Proof of Upper Bound:* In the best-case scenario, where each split level appears exactly once in the decision tree (i.e., there is no redundancy among the tree nodes), the omniscient algorithm (a.k.a optimal algorithm) would require at least $n + 1$ queries to reconstruct the tree:

$$Q_{opt}^f \geq n + 1 \qquad (2)$$

This includes one query for each leaf to verify its label and certify functional equivalence. Note that this assumes that the omniscient algorithm has correctly guessed the location of each split level and directly queried for counterfactuals over each leaf region.

Conversely, in the worst-case scenario, such as a chessboard-like decision tree where splits are evenly distributed across multiple features, the TRA algorithm must explore all possible regions created by these splits. For a two-dimensional tree, this results in $s_1 + s_2(s_1 + 1) + (s_1 + 1)(s_2 + 1)$ queries, where $s_1$ and $s_2$ are the number of splits along each feature. The $s_1$ splits along the first dimension are first detected, then the $s_2$ splits along the second dimension are re-discovered at every sub-division performed along the first dimension, and finally $(s_1 + 1)(s_2 + 1)$ queries are required to individually verify each sub-square (leaf node). Extending this to $m$ dimensions, the number of queries grows multiplicatively with the number of splits per feature, leading to:

$$Q_{TRA}^f \leq \sum_{i=1}^{m} s_i \prod_{j=1}^{i-1} (s_j + 1) + \prod_{j=1}^{m} (s_j + 1) = 2 \prod_{j=1}^{m} (s_j + 1) - 1$$

Therefore, the competitive ratio $C_{TRA}^{(n,m)}$ is bounded above by:

$$C_{TRA}^{(n,m)} = \sup_{f \in \mathcal{F}} \left( \frac{Q_{TRA}^f}{Q_{opt}^f} \right) \leq \frac{2 \prod_{j=1}^{m} (s_j + 1) - 1}{n + 1} \qquad (3)$$

2. *Proof of Lower Bound:* We hereafter build an *adversarial example*, i.e., one that maximizes the ratio of the number of queries that TRA must perform to extract the target decision tree, compared to what an optimal offline algorithm could achieve. This example therefore constitutes a (feasible) lower bound for the competitive ratio of TRA.

Consider a tree with $n$ splits. An adversary (dynamically building the worst-case instance the online algorithm is run on) can arrange the splits such that the first split detected by TRA is the last decision node in the tree. Specifically, the adversary ensures that the initial split does not reduce the complexity of identifying the remaining $n$ splits.

Consider the following adversarial example: for each $1 \leq p \leq n$ and $1 \leq j \leq m$, let the dimension that $\alpha_p$ splits on be $j$, and set

$$\alpha_p = \begin{cases} \frac{p}{(s_1+1)}, & \text{if } j = 1, \\ \frac{p - \sum_{i=1}^{j-1} s_i}{2(s_j+1)} + \frac{1}{2} + \epsilon, & \text{otherwise,} \end{cases}$$

where $\epsilon > 0$. This adversarial example ensures that within any hyper-rectangle defined by split level boundaries, for $j = 1, ..., m - 1$ if there are splits in both the $j$-th and $j + 1$-th dimensions, then there exists a split in the $j$-th dimension that is closer to the center of the hyper-rectangle than any split in the $j + 1$-th dimension. As a consequence, TRA will always detect the splits of the $j$-th dimension before those of the $j + 1$-th dimension. Therefore,

the adversary can design a decision tree with a single branch (as illustrated in the right tree of Figure 7) that begins by splitting on the split levels in reverse (decreasing) order of dimensions (see a 2D example in Figure 7). For this specific example, TRA will require

$$2 \prod_{j=1}^{m} (s_j + 1) - 1$$

queries, whereas the optimal offline algorithm only needs $n + 1$ queries.

Therefore, by the definition of competitive ratio:

$$C_{TRA}^{(n,m)} \geq \frac{2 \prod_{j=1}^{m} (s_j + 1) - 1}{n + 1} \tag{4}$$

Hence, by (3) and (4), we have:

$$C_{TRA}^{(n,m)} = \frac{2 \prod_{j=1}^{m} (s_j + 1) - 1}{n + 1}$$

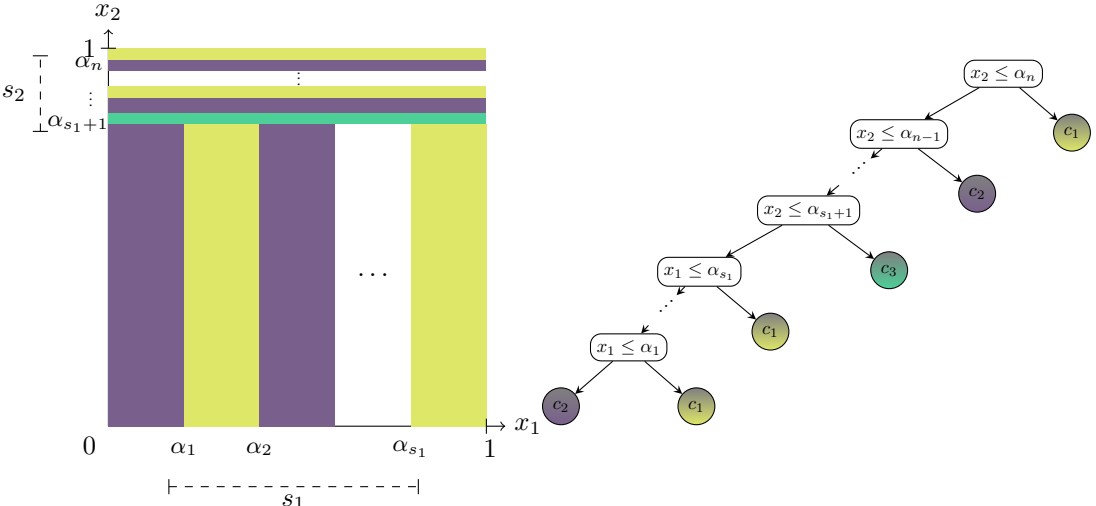

Figure 7: Adversarial example for TRA (displayed for $m = 2$ dimensions). The classes are $c_1$, $c_2$ and $c_3$. For simplicity, we denote $s_1 = s_1^{(n)}$ and $s_2 = s_2^{(n)}$. Here, the instance is dynamically built so that the number of queries required by TRA is $s_2(s_1 + 1) + s_1 + (s_1 + 1)(s_2 + 1)$, whereas the optimal offline algorithm only needs $n + 1$ queries to check the leaf labels and certify functional equivalence, as shown in the right tree figure.

$\square$

**Proposition 3.7.** *For all $n > 0$ and $m \geq 2$, no divide-and-conquer-based algorithm can achieve a competitive ratio better than $C_{TRA}^{(n,m)}$.*

*Proof of Proposition 3.7.* **Key Idea.** A pure divide-and-conquer (D&C) algorithm discovers a split along a specific feature dimension upon querying a point. This split divides the input space into two subproblems. An adversary can strategically arrange the splits so that the feature splits detected by the algorithm early on are the "least helpful" ones, meaning they occur as the last decisions along their respective feature branches. By doing this, the adversary ensures that these initial splits do not simplify the identification of the remaining splits. We demonstrate that this construction forces the D&C algorithm to perform poorly compared to an optimal strategy.

We define a pure D&C-based algorithm as one that divides the input space (problem) into subspaces (sub-problems) based on counterfactual explanations and recursively continues this process within each subspace until no counterfactuals are found. This class of algorithms encompasses all types of

querying strategies, such as selecting the geometrical center (as done by TRA), but also selecting the top-left corner, bottom-right corner, or a random point within the input space, among others.

**Proof.** Let $m > 1$ be the number of dimensions and $n \geq m$ be the number of split levels. We prove this proposition by induction on the number of split levels $n > 0$.

- **Base Case** ($n = 2$, $m = 2$): Consider a two-dimensional tree with split levels $\alpha_1$ and $\alpha_2$. Let $q = (q_1, q_2)$ be the query made by the D&C algorithm. The adversary chooses $\alpha_1 = q_1 + \epsilon_1$ and $\alpha_2 = q_2 + \epsilon_2$, where $\epsilon_2 > \epsilon_1 > 0$. Consequently, the first counterfactual explanation returned by the oracle is $q' = (\alpha_1, q_2)$. The algorithm then splits the input space into two subspaces, both containing the split at $\alpha_2$, as depicted in Figure 8.

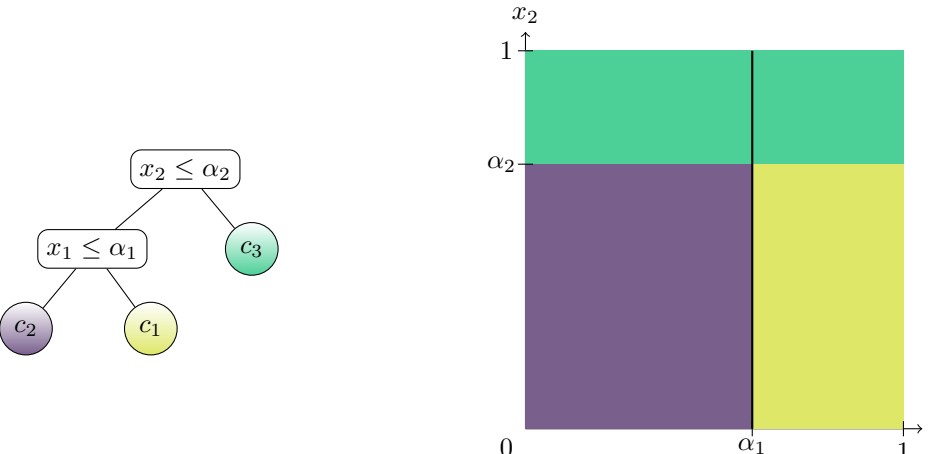

(a) Adversarial decision tree.     (b) Decision Boundary of the adversarial decision tree.

Figure 8: An adversarial example for $n = 2$, $m = 2$, triggering the worst-case competitive ratio of our algorithm.

- In this adversarial example, the D&C algorithm requires at least 7 queries to reconstruct the exact decision tree, whereas an omniscient optimal algorithm can achieve this with only 3 queries, one per leaf. Therefore, for this adversarial example, no D&C-based algorithm can attain a competitive ratio better than $C_{TRA}^{n,m} = C_{TRA}^{2,2} = \frac{7}{3}$.

- **Inductive Step**: Assume the proposition holds for all trees with up to $n$ split levels. Consider a tree with $n+1$ split levels, with split levels $(\alpha_1, \ldots, \alpha_{n+1})$. Let $q = (q_1, q_2, \ldots, q_m) \in \mathcal{X}$ be the first query made by the D&C algorithm. The adversary sets $\alpha_1 = q_1 + \epsilon_1$ where $\epsilon_1 > 0$ and returns the counterfactual explanation $q' = (\alpha_1, q_2, \ldots, q_m)$. The adversary places this split as the last decision node in the tree. Consequently, the D&C algorithm splits the input space into two subspaces, each containing all splits of the remaining dimensions, thereby containing at most $n$ splits each.

By the induction hypothesis, the algorithm will require at least:

$$Q_1 = 2(s_1^{(1)} + 1) \prod_{j=2}^{m} (s_j + 1) - 1$$

for the first subspace, and

$$Q_2 = 2(s_1^{(2)} + 1) \prod_{j=2}^{m} (s_j + 1) - 1$$

for the second subspace, where $s_1^{(1)}$ and $s_1^{(2)}$ are the remaining splits along the first dimension in the first and second subspaces, respectively.

Therefore, the total number of queries is:

$$1 + Q_1 + Q_2 = 2 \prod_{j=1}^{m}(s_j + 1) - 1 = (n+1)C_{TRA}^{(n,m)} \qquad (5)$$

given that $s_1^{(1)} + s_1^{(2)} + 1 = s_1$. Hence, by (2) and (5), the best competitive ratio $C_{D\&C}^{(n,m)}$ achievable by any D&C-based algorithm satisfies:

$$C_{D\&C}^{(n,m)} \geq \frac{(n+1)C_{TRA}^{(n,m)}}{n+1} = C_{TRA}^{(n,m)}.$$

$\square$

# B    Details of the SPLIT Procedure (Used by TRA)

Given a region $\mathcal{E}$, a query $x$, and its counterfactual $x'$, SPLIT partitions $\mathcal{E}$ into disjoint subregions by iterating over features where $x$ and $x'$ differ, peeling off the half that contains $x$ and keeping the complementary half (the side toward $x'$). The exact pseudo-code of this procedure is provided in Algorithm (2).

---

**Algorithm 2** SPLIT$(\mathcal{E}, x, x')$

---

**Require:** Region $\mathcal{E}$; vectors $x, x'$.
**Ensure:** List $E$ of disjoint subregions whose union is $\mathcal{E}$.
 1: $S \leftarrow \{ (i, x_i') \mid x_i \neq x_i' \}$ {indices and thresholds where $x$ and $x'$ differ}
 2: $\mathcal{E}_0, E \leftarrow \mathcal{E}, \emptyset$
 3: **for** $(i, v) \in S$ **do**
 4:     **if** $x_i \leq v$ **then**
 5:         $\mathcal{E}_1 \leftarrow \{ z \in \mathcal{E}_0 \mid z_i \leq v \}$
 6:     **else**
 7:         $\mathcal{E}_1 \leftarrow \{ z \in \mathcal{E}_0 \mid z_i > v \}$
 8:     **end if**
 9:     $E \leftarrow E \cup \{\mathcal{E}_1\}$ {peel off the side containing $x$}
10:     $\mathcal{E}_0 \leftarrow \mathcal{E}_0 \setminus \mathcal{E}_1$ {keep the remainder (toward $x'$)}
11: **end for**
12: $E \leftarrow E \cup \{\mathcal{E}_0\}$ {add the final remainder}
13: **return** $E$

---

**Tie-handling.**    To avoid overlaps at $v$, we use a consistent rule (e.g., $\leq$ on one side and $>$ on the other). In numerical implementations, we replace the strict inequality ($z_i > v$) with a relaxed one $z_i \geq v + \varepsilon$ for a small $\varepsilon > 0$.

# C    Additional Experimental Results

## C.1    Experimental Setup Details

**Target Model Training.**    During the training process of both decision trees and random forests, we conduct a grid search with 50 steps over the range $[0, 0.2]$ to determine the optimal cost-complexity pruning parameter `ccp_alpha` that maximizes accuracy on the validation dataset.

**Surrogate Model Training.**    For surrogate models (used by the CF and DUALCF attacks) that do not utilize the target model's hyperparameters, we employ the default parameters provided by the scikit-learn Python library [Pedregosa et al., 2011]. Specifically for MLPs, we configure a scikit-learn MLP with two hidden layers, each consisting of 20 neurons, while keeping all other parameters at their default values.

**Anytime Fidelity.** The anytime fidelity was calculated each 20 queries during all attacks execution, except for PATHFINDING which is not an anytime attack. Let $N$ denote the number of target models $f_1, f_2, \ldots, f_N$ to extract (i.e., over all considered experimental configurations and random seeds) and $\mathcal{D}$ a dataset with $n$ samples. The anytime fidelity of a given model extraction attack (at a given time step) over $\mathcal{D}$ is calculated as follows :

$$\frac{1}{N} \sum_{i=1}^{N} \left( \frac{1}{n} \sum_{j=1}^{n} \mathbb{1}_{\{f_i(x_j) = \hat{f}_i(x_j)\}} \right) \tag{6}$$

where $\hat{f}_1, \hat{f}_2, \ldots, \hat{f}_N$ are the models extracted by the considered model extraction attack at the given time step.

### C.2 Configuration of Surrogate-Based Attacks

We report in Figure 9 (respectively, Figure 10) the anytime performance of the CF (respectively, DUALCF) model extraction attack against decision tree models, for the three considered types of surrogates and the two counterfactual oracles, on all considered datasets. More precisely, as depicted in Section 4.1, we run these two attacks using three different assumptions on adversarial knowledge, namely the hypothesis class of the target model, its hyperparameters, and none of them. In the first case, the adversary trains a decision tree surrogate with default parameters (DT). In the second case, he trains a surrogate decision tree with the exact same hyperparameters as the target model (DT+). Finally, in the third case, a multi-layer perceptron (MLP) is used as surrogate model. Both CF and DUALCF were originally tested using the DiCE [Mothilal et al., 2020] counterfactual oracle, which provides heuristic-based (non-optimal) counterfactual explanations. To assess the impact of explanation optimality on the attack's performance, and to ensure fair comparisons, we run CF and DUALCF both using DiCE and using optimal counterfactual explanations computed with the OCEAN framework [Parmentier and Vidal, 2021].

We also report in Figure 11 the anytime performance of the CF and DUALCF model extraction attacks against random forest models, for the three considered types of surrogates and the two counterfactual oracles, on the COMPAS dataset.

We hereafter highlight the key trends of these results, focusing on the impact of two dimensions: the adversarial knowledge (regarding the target model's architecture and hyperparameters) and the type of counterfactual oracle used.

**Knowledge of the target model architecture and hyperparameters.** One first important trend that is consistent across both CF and DUALCF, and for both decision trees and random forests, is that knowledge of the hypothesis class of the target model helps fitting a surrogate with high fidelity. Indeed, as can be observed in Figures 9, 10 and 11, the MLP surrogate always under-performs compared to the decision trees or random forests ones. Indeed, fitting a surrogate model of the same type is facilitated by the fact that the shapes of its decision boundaries are the same as the target model, e.g., axis-parallel splits for tree-based models. Interestingly, knowledge of the hyperparameters of the target decision tree or random forests does not seem to help fitting the surrogate. Indeed, in most experiments, the surrogate sharing the same hypothesis class as the target model (i.e., DT or RF) and the surrogate sharing both the hypothesis class and the hyperparameters (i.e., DT+ or RF+) have very close performances. Furthermore, imposing the target model's hyperparameters to the trained surrogate can even be counterproductive, as can be seen in Figures 9a and 9b for instance. In such cases, the fact that surrogate learning is more constrained due to the enforced hyperparameters (e.g., maximum depth) seems to slow its convergence towards very high fidelity values. This is consistent with previous findings: for instance, Aïvodji et al. [2020] observed that knowledge of the architecture of a target neural network did not provide a significant advantage to the CF model extraction attack.

**Optimality of the counterfactuals.** The results in Figures 9, 10 and 11 suggest that the non-optimal counterfactual explanations returned by DiCE helped fitting the extracted surrogate models better than the optimal ones provided by OCEAN. Indeed, for a fixed query budget and surrogate type, the performances of the extracted model are often better when fitted with DiCE counterfactuals than with OCEAN ones. Although some variations appear, this finding is generally verified for all the studied types of surrogates, for both decision trees and random forests target models, and for both the

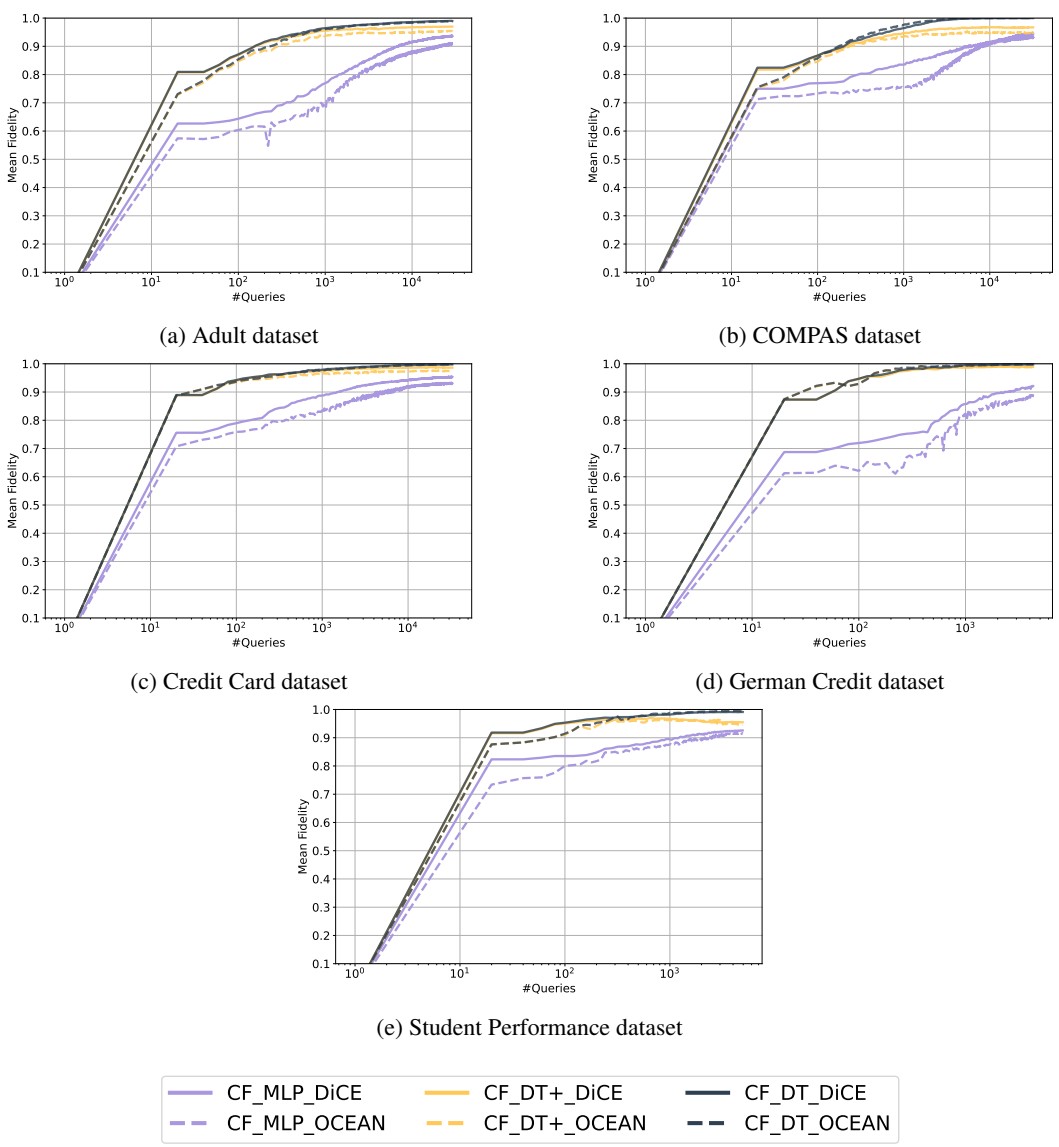

(a) Adult dataset  (b) COMPAS dataset

(c) Credit Card dataset  (d) German Credit dataset

(e) Student Performance dataset

Figure 9: Anytime performance of the CF model extraction attack against decision trees. We report results for all datasets and studied configurations, including adversarial knowledge regarding the target model architecture (DT, DT+, and MLP) and counterfactual oracles (DiCE and OCEAN).

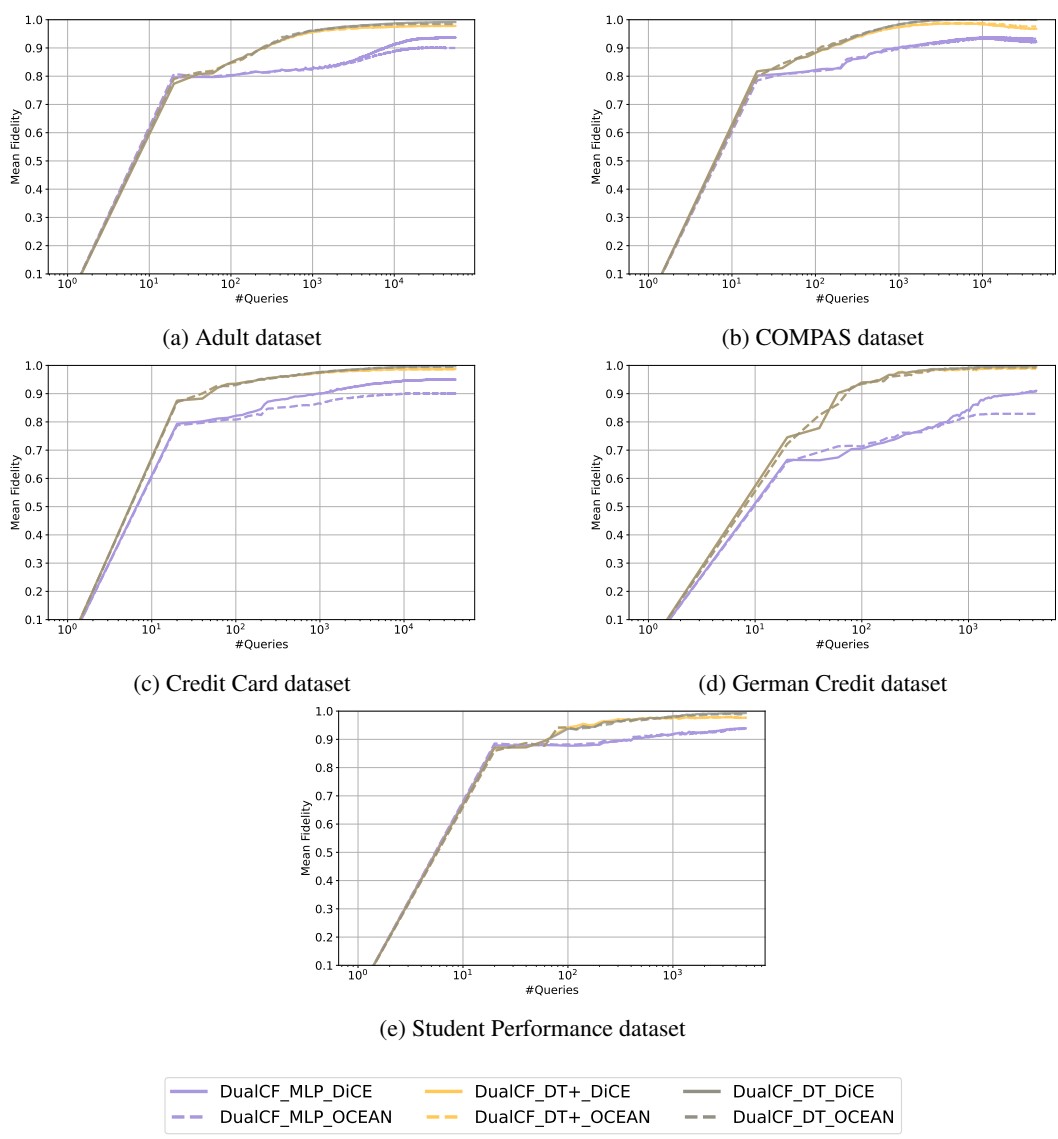

Figure 10: Anytime performance of the DUALCF model extraction attack against decision trees. We report results for all datasets and studied configurations, including adversarial knowledge regarding the target model architecture (DT, DT+, and MLP) and counterfactual oracles (DiCE and OCEAN).

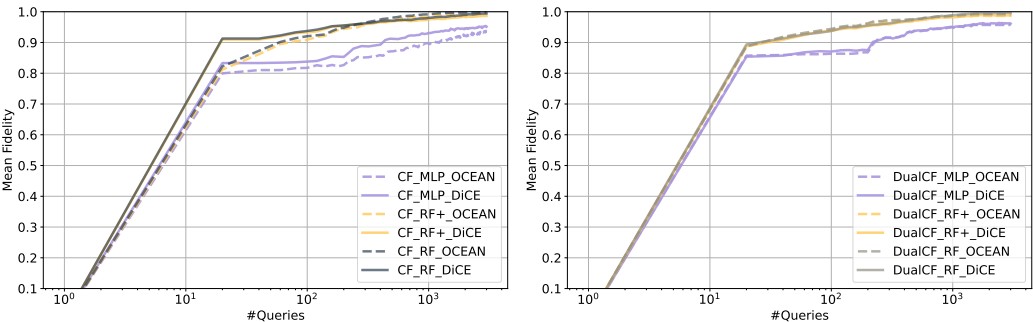

Figure 11: Anytime performance of the CF (left) and DUALCF (right) model extraction attacks against random forests. We report results for the COMPAS dataset and all studied configurations, including adversarial knowledge regarding the target model architecture (DT, DT+, and MLP) and counterfactual oracles (DiCE and OCEAN).

CF and DUALCF extraction attacks. Intuitively, this can be attributed to a greater diversity in the non-optimal counterfactuals, which do not necessarily lie close to a decision boundary, unlike optimal ones. This also highlights a crucial insight: optimal counterfactuals only give an advantage to a model extraction attack if the attack is able to leverage the information it carries as a whole (including both the counterfactual example and its optimality) through a structured approach, as demonstrated by TRA.

### C.3 Test Set Fidelity Results

As mentioned in Section 4.1, the results provided in Section 4.2 measure fidelity on a dataset uniformly sampled over the input space, which accurately quantifies how well the extracted models fit the decision boundaries of the target ones over the whole input space. Another approach consists in evaluating fidelity on a test set. In such cases, the results indicate how well the extracted models mimic the target ones for examples drawn from the actual data distribution. We report in Table 3 the results of our extraction attacks against random forests. More precisely, for random forests of varying sizes, we report the average fidelity (measured on the uniformly sampled dataset or on a test set) achieved by all considered methods along with the required number of queries. For DUALCF and CF, these values are arbitrarily fixed to allow their surrogates to converge towards (near) perfect fidelity. For our proposed TRA, functional equivalence is achieved using the reported number of queries, hence fidelity on both considered datasets is always $1.0$. For CF and DUALCF, we report results for the random forest surrogate using default parameters, for both studied counterfactual oracles. Indeed, we observed in Section C.2 that DiCE counterfactuals led to better anytime performances (in terms of uniform dataset fidelity) than OCEAN ones within the CF model extraction attack, in most experiments. However, this is not always the case, with a few setups in which the difference between both approaches becomes very small or shifts in favor of the runs using OCEAN counterfactuals after sufficiently many iterations. This is the case on the COMPAS dataset (Figure 9b), and although the difference remains very subtle, it is also visible on the test set fidelity, illustrating the fact that the two values are often very well aligned.

Overall, the superiority of TRA is clear, both in terms of (uniform or test) fidelity and in terms of required numbers of queries, confirming the observations of Section 4.2.

Table 3: Summary of our model extraction experiments against random forests, on the COMPAS dataset. For random forests with varying numbers of trees, we report their total number of nodes and the average performances of the different considered model extraction attacks. FU and FTD denote respectively the Fidelity over the Uniform and Test Data.

| | | | TRA | | | DualCF | | | | | | CF | | | | | |
| | | | | | | | | | RF | | | | | | RF | | |
| | | | OCEAN | | | DiCE | | | OCEAN | | | DiCE | | | OCEAN | | |
| Dataset | #Trees | #Nodes | #Queries | FU | FTD | #Queries | FU | FTD | #Queries | FU | FTD | #Queries | FU | FTD | #Queries | FU | FTD |
|---|---|---|---|---|---|---|---|---|---|---|---|---|---|---|---|---|---|
| COMPAS | 5 | 486.60 | 73.60 | 1.00 | 1.00 | 3000 | 1.00 | 0.99 | 3000 | 1.00 | 1.00 | 3000 | 1.00 | 1.00 | 3000 | 1.00 | 1.00 |
| | 25 | 4569.00 | 138.80 | 1.00 | 1.00 | 3000 | 0.99 | 0.98 | 3000 | 1.00 | 1.00 | 3000 | 1.00 | 1.00 | 3000 | 1.00 | 1.00 |
| | 50 | 9151.20 | 147.60 | 1.00 | 1.00 | 3000 | 0.99 | 0.98 | 3000 | 1.00 | 1.00 | 3000 | 1.00 | 1.00 | 3000 | 1.00 | 1.00 |
| | 75 | 7317.00 | 95.20 | 1.00 | 1.00 | 3000 | 1.00 | 0.99 | 3000 | 1.00 | 1.00 | 3000 | 1.00 | 1.00 | 3000 | 1.00 | 1.00 |
| | 100 | 18369.20 | 129.60 | 1.00 | 1.00 | 3000 | 1.00 | 0.98 | 3000 | 1.00 | 1.00 | 3000 | 1.00 | 1.00 | 3000 | 1.00 | 1.00 |

## C.4 Detailed Experimental Results

We hereafter report all the results of our main experiments over the five considered datasets.

First, Figure 12 provides the anytime performances (in terms of average surrogate fidelity as a function of the number of performed queries) of the four considered model extraction attacks when applied on decision tree target models. The findings highlighted in Section 4.2 (in particular, **Result 1**) are consistent across all considered datasets: TRA exhibits higher anytime fidelity than CF and DUALCF for all query budgets, while also providing functional equivalence guarantees. While PATHFINDING also provides these guarantees, it necessitates orders of magnitudes more queries to fit its surrogate.

Figure 13 focuses on functionally equivalent model extraction attacks, and relates the number of queries they require to fully recover the target model to its size (quantified as its number of nodes). The logarithmic scale of the y-axis highlights that TRA usually requires orders of magnitudes fewer queries than PATHFINDING to entirely extract a given decision tree, confirming our **Result 2** (Section 4.2). Interestingly, this trend is more subtle when reconstructing large trees trained on the datasets with the highest numbers of features (i.e., Adult and SPerformance).

Finally, Figure 14 reports the number of counterfactual queries required by TRA to conduct a functionally equivalent extraction of random forests of various sizes, as a function of the total number of nodes to be recovered within the target forest. Importantly, as quantified through the performed power-law regression, the number of queries required by TRA to entirely extract the target forests grows sub-linearly – in $\Theta(\#\mathrm{Nodes}^{0.38})$ – with the total number of nodes to be retrieved. This empirically demonstrates the very good scalability of TRA with respect to the size of the target random forests, consistent with **Result 3** (Section 4.2). Note that this behavior arises because large forests with many trees introduce redundancies, allowing the extracted model to be represented with perfect fidelity as a more compact decision tree [Vidal and Schiffer, 2020].

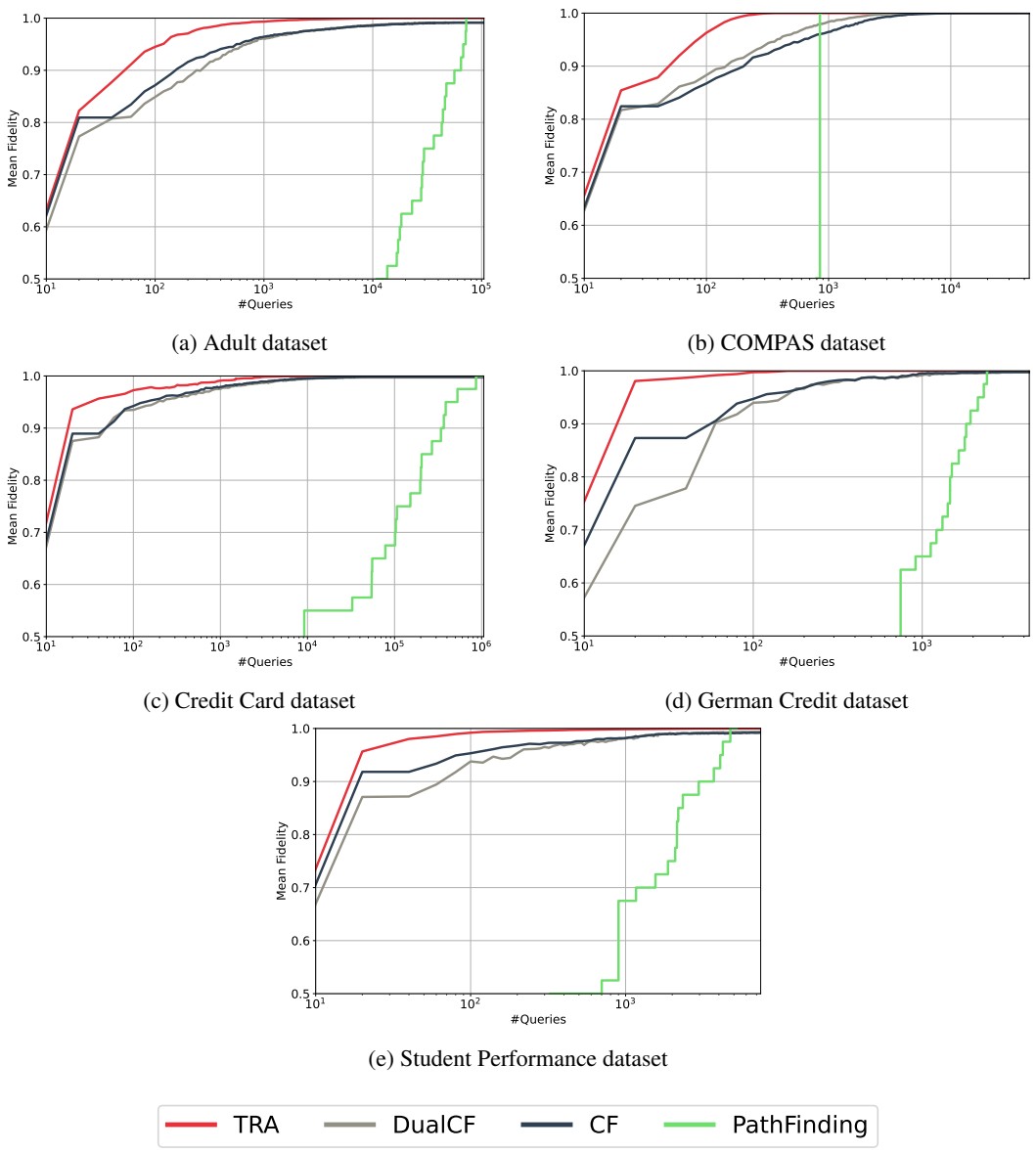

Figure 12: Anytime performance of all the considered model extraction attacks against decision trees. We report results for all datasets and retain the best configuration for the surrogate-based attacks CF and DUALCF.

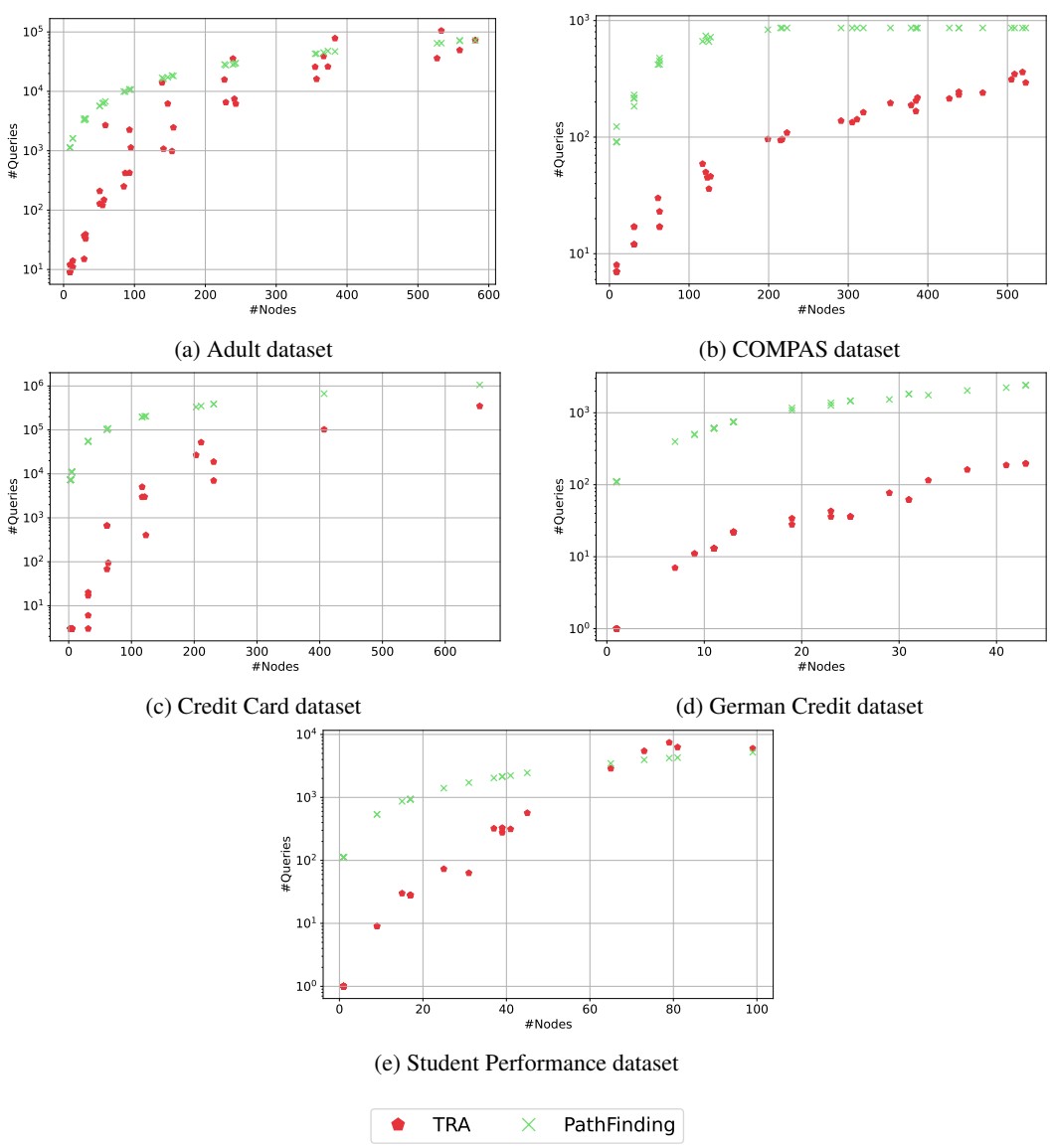

Figure 13: Performance of the PATHFINDING and TRA functionally equivalent model extraction attacks against decision trees. We report results for all datasets where each point represents the number of queries needed to fully reconstruct the trees.

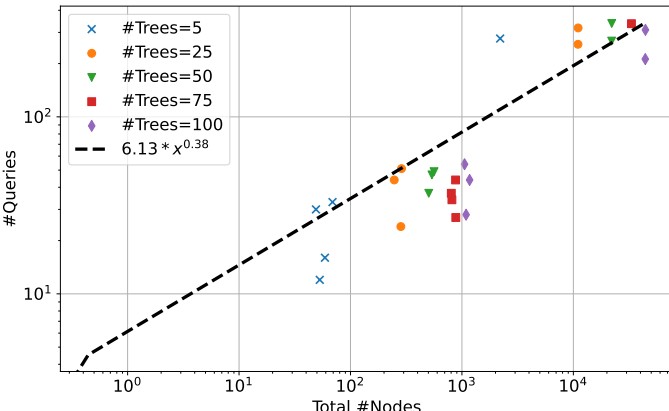

Figure 14: Number of queries required by TRA to perform a functionally equivalent extraction of target random forests of various sizes on the COMPAS dataset, as a function of the total number of nodes to be reconstructed within the target forest. As illustrated through the performed power-law regression, the number of required queries grows sub-linearly – in $\Theta(\#\text{Nodes}^{0.38})$ – with the total number of nodes to be retrieved, suggesting good scalability of the extraction attack with respect to the forest size.

# D    Performances of TRA when used with Locally Optimal Counterfactuals

In the main body of the paper (Section 4.2) we employed the OCEAN counterfactual oracle for our experiments using TRA. It supplies *globally* optimal counterfactual explanations. While such global optimality is *sufficient* for the TRA attack to succeed, it is *not necessary*. Indeed, as pointed out in Section 3.2, TRA requires counterfactuals that are *locally* optimal – i.e., positioned on (or infinitesimally close to) the decision boundary of the explained model $f$. To illustrate this distinction, we replicate all experiments, following exactly the same setup described in Section 4.1, using locally optimal counterfactuals.

The DiCE counterfactual oracle, used in our experiments for the CF and DUALCF model extraction attacks, could be a good candidate. However, it frequently fails to return valid counterfactuals when restricted to a given subspace of the input space – making it difficult to use within TRA. This suggests that the counterfactuals found by DiCE often lie in different decision regions of the target model, rather far from the actual query. When the counterfactuals are used to provide recourse information, this can be problematic – for instance, by suggesting that a credit applicant must exert significantly more effort than actually necessary to improve their application. For these reasons, in our experiments using TRA, we rather consider a lightweight heuristic (Algorithm 3) producing locally optimal counterfactuals.

---

**Algorithm 3** A simple heuristic to find locally optimal counterfactual explanations

---

1: **Input:** Query $x$, model $f$, input space $\mathcal{E}$, training data $\mathcal{D}_T$, and maximum number of iterations for the uniform sampling process $S$.
2: **Return:** A locally optimal counterfactual explanation $x' \in \mathcal{E}$ if it exists.
3: $\mathcal{D}_{\mathcal{E}} \leftarrow \mathcal{D}_T \cap \mathcal{E}$ {gets all the data points that are in the desired input space $\mathcal{E}$}
4: **for** $x' \in \mathcal{D}_{\mathcal{E}}$ **do**
5:    **if** $f(x') \neq f(x)$ **then**
6:       **return** $linesearch(x, x')$ {$x'$ is a counterfactual, we refine it via line search towards $x$}
7:    **end if**
8: **end for**
9: **for** $0 \leq i \leq S$ **do**
10:    $x' \leftarrow sample(\mathcal{E})$ {sample a point $x'$ uniformly within $\mathcal{E}$}
11:    **if** $f(x') \neq f(x)$ **then**
12:       **return** $linesearch(x, x')$ {$x'$ is a counterfactual, we refine it via line search towards $x$}
13:    **end if**
14: **end for**
15: **return** $None$ {no counterfactual found}

---

More precisely, Algorithm 3 searches in the training dataset and then refines the first valid counterfactual it finds via a one-dimensional line search (line 6). If no counterfactual is found, it tries another strategy by sampling a predefined number of points uniformly in $\mathcal{E}$. In our experiments, the maximum number of such sampled points is fixed to $S = 1000$. If a valid counterfactual is found, it is refined via line search and returned (line 12). Because the search is stochastic and local, it may fail to return a counterfactual even when one exists, yet it satisfies local optimality whenever a solution is found.

Figures 15 and 16 contrast the performance of TRA to extract decision trees, using OCEAN (globally optimal counterfactuals) versus Algorithm 3 (simpler algorithm, producing locally optimal counterfactuals) as counterfactual oracle. The observed trends confirm that TRA remains effective as long as the returned counterfactuals are locally, though not necessarily globally, optimal – thereby confirming **Result 4** (Section 4.2). Interestingly, we observe in Figure 15 that in the early iterations, locally optimal counterfactuals may lead to better anytime fidelity values – which is likely the case due to their heuristic nature, enhancing diversity. We also observe a pathological case in Figure 15a, with the Adult dataset. In this particular experiment, when used with Algorithm 3 as counterfactual oracle, TRA converges close to near-perfect – but not exact – fidelity. This is due to the fact that Algorithm 3 fails to find feasible counterfactuals within the desired region, even if one exists. Indeed, even if the counterfactuals provided by this simple oracle satisfy local optimality whenever one is found, there is no guarantee that if none is found that no one actually exists. Performing a finer uniform sampling (via raising the number of sampled points $S$) could fix this issue. Nevertheless, the

performances of TRA with this weak oracle remain very robust, as evidenced through our large set of experiments.

The performances of TRA to extract random forests target models, with either the OCEAN or Algorithm 3 counterfactual oracles, are provided in Table 4 and display the same trends. TRA is able to achieve perfect fidelity for both counterfactual oracles, and there is no significant difference in the required number of queries (which remains very low compared to the size of the reconstructed forests, as discussed in Section 4.2).

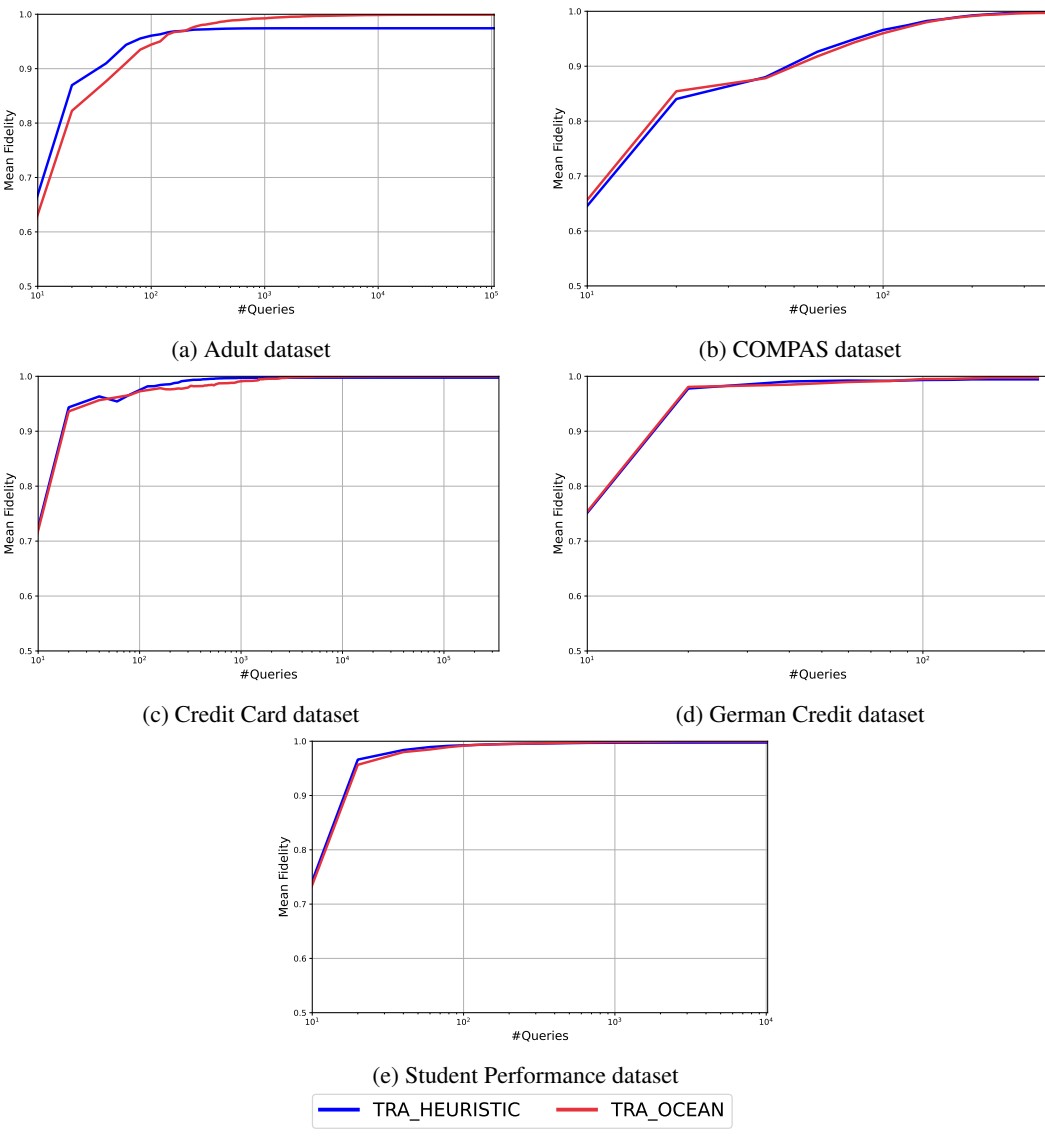

Figure 15: Anytime performance of TRA with either OCEAN or a simpler heuristic counterfactual oracle (Algorithm 3) to extract decision trees. We report results for all datasets.

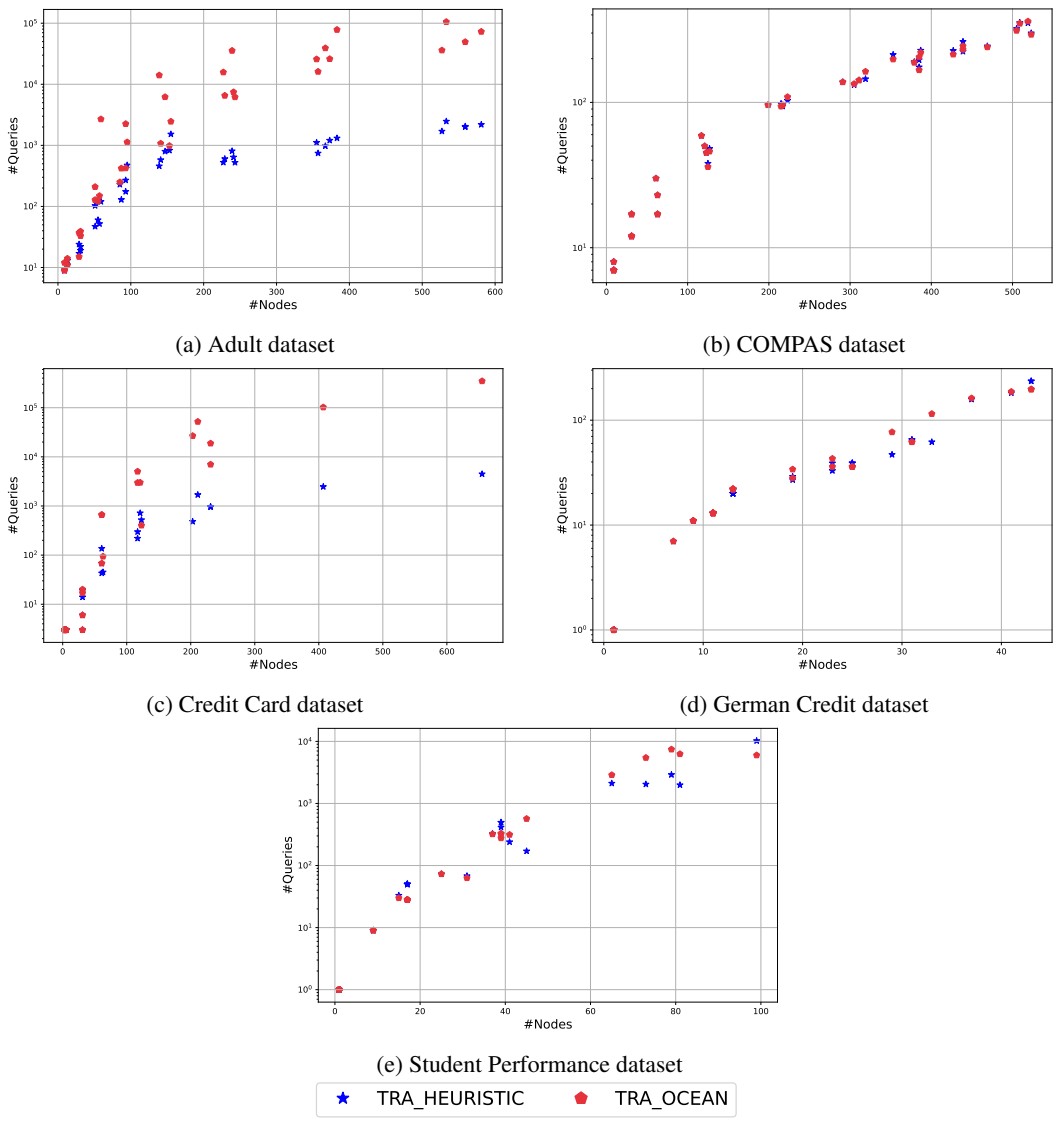

(a) Adult dataset

(b) COMPAS dataset

(c) Credit Card dataset

(d) German Credit dataset

(e) Student Performance dataset

★ TRA_HEURISTIC     ⬟ TRA_OCEAN

Figure 16: Performance of TRA with either OCEAN or a simpler heuristic counterfactual oracle (Algorithm 3) to achieve functionally equivalent model extraction attacks against decision trees. We report results for all datasets where each point represents the number of queries needed to fully reconstruct the trees.

Table 4: Summary of our model extraction experiments against random forests, on the COMPAS dataset using both OCEAN and Heuristic. FU and FTD denote respectively the Fidelity over the Uniform and Test Data.

| | | | TRA | | | | | |
|---|---|---|---|---|---|---|---|---|
| | | | OCEAN | | | Heuristic | | |
| Dataset | #Trees | Nodes | #Queries | FU | FTD | #Queries | FU | FTD |
| COMPAS | 5 | 486.60 | 73.60 | 1.00 | 1.00 | 74.20 | 1.00 | 1.00 |
| | 25 | 4569.00 | 138.80 | 1.00 | 1.00 | 140.00 | 1.00 | 1.00 |
| | 50 | 9151.20 | 147.60 | 1.00 | 1.00 | 149.20 | 1.00 | 1.00 |
| | 75 | 7317.00 | 95.20 | 1.00 | 1.00 | 95.20 | 1.00 | 1.00 |
| | 100 | 18369.20 | 129.60 | 1.00 | 1.00 | 130.40 | 1.00 | 1.00 |

# E   Broader Impact Statement

To meet ethical and legal transparency requirements, machine learning explainability techniques have been extensively studied in recent years. Among them, counterfactual explanations provide a natural and effective approach by identifying how an instance could be modified to receive a different classification. In credit granting applications, for example, they can provide recourse to individuals whose credit was denied.

As a result, MLaaS platforms increasingly integrate such explainability tools into their APIs. While these explanations enhance user trust, they also expose a new attack surface to malicious entities by revealing additional model information. In this work, we theoretically and empirically demonstrate that locally optimal counterfactual explanations of decision trees and tree ensembles can be exploited to conduct efficient model extraction attacks. These results highlight the critical tension between transparency and the protection of model integrity and intellectual property. By identifying and quantifying these vulnerabilities, we highlight the risks of releasing model explanations without a thorough security assessment. Our research establishes a benchmark for evaluating method safety, advocating for the development of privacy-preserving approaches to explainability.

Finally, while previous evaluations of model extraction attacks have been predominantly empirical, we show that tools from online discovery provide a principled framework for characterizing attack efficiency. This perspective paves the way for more structured approaches to assessing model attack budgets and risks.

