# OpenReview forum: "From Counterfactuals to Trees: Competitive Analysis of Model Extraction Attacks"
_NeurIPS.cc/2025/Conference — NeurIPS 2025 spotlight_

### Official Review · Reviewer_hAR3 · 2025-06-25

**Clarity:** 4
**Significance:** 3
**Originality:** 3
**Rating:** 5
**Confidence:** 3

**Summary:**

In this paper, the authors present a framework to understand model extraction attacks. They also introduce a new algorithm for mode extraction attacks. They suggest using the competitive ratio to analyze extraction attacks as their framework. They prove their algorithm is optimal in the number of queries with respect to the competitive ratio among divide-and-conquer algorithms.

**Questions:**

Yes, I am not very familiar with the area, and I come from algorithm analysis, so I have several questions:
1. The competitive ratio in the algorithmic sense is based on the worst-case instance, while more modern literature, such as algorithms with predictions, sometimes uses the average case. In Definition 3.3, is the minimal number of queries for the worst-case instance?
2. What is the analysis of the competitive ratios of the existing baselines (e.g., Pathfinding, CF, Dual CF): are your empirical results showing something or just validating your theoretical results?
3. $m$, where most of the features are one-hot encoding vs $m$ when most features are continuous, seems like different levels of difficulty. I see Table 1 separates m into different features. How does this impact your analysis? Proposition 3.6 seems like a loose upper bound if all m features are one-hot encoding?
4. What happens when the outputs of your oracle are noisy? E.g., if the counterfactual explanation is wrong with some probability.

**Ethical Concerns:**

["NO or VERY MINOR ethics concerns only"]

**Final Justification:**

The authors sufficiently address my concerns. I will raise the score to accept. But I want to reiterate that I am not an expert in the area and defer to other reviewers for final decisions.

**Limitations:**

Can the authors point me to their limitations section?

**Quality:**

3

**Strengths And Weaknesses:**

Strengths:
1. Using the competitive ratio to analyze extraction attacks seems pretty natural. I am glad the authors suggested it.
2. The authors provide empirical results across multiple datasets.
3. The paper is high quality, and the proofs are detailed and easy to follow.
4. The paper is clearly written, and the results are well communicated. Algorithms are clearly described and the empirical experiments are well explained.

Weaknesses:
1. Motivation for the theoretical framework can be better described.
2. Connections between theoretical results and empirical results can be strengthened.

I am not familiar enough with the field to comment on Originality and Clarity.

---

> ### Author Rebuttal · Authors · 2025-07-26
>
> Thank you for your insightful comments.
>
> ---
> > Motivation for the theoretical framework can be better described.
>
> Sure, we will strengthen the introduction by explicitly motivating our use of competitive analysis: it provides a rigorous, model-agnostic way to quantify the hardness of tree extraction attacks and bound the number of queries. This will clarify why our theoretical framework is both natural and important for this problem.
>
> ---
> >  Connections between theoretical results and empirical results can be strengthened.
>
> Thank you for this suggestion, it aligns with a similar comment from Reviewer oRDk. In our revised manuscript, we will include a new table in Appendix C and reference it in Section 4.2. This table directly compares the **empirical** number of queries required by TRA to achieve functional equivalence against the **theoretical** worst-case bound derived from Proposition 3.6 on each dataset. This comparative view shows that, in practice, TRA is significantly more efficient than the pessimistic theoretical bound:
> |Dataset | Empirical \# queries | Worst-case \# queries|
> |--|:--:|:--:|
> |SPerformance | $1.16e+03$ | $3.02e+08$|
> |Adult | $3.70e+04$ | $2.45e+14$|
> |German Credit | $5.18e+01$ | $2.86e+03$|
> |Credit Card | $6.97e+04$ | $6.49e+11$|
> |COMPAS | $1.53e+02$ | $1.07e+03$ |
>
> ---
> >  The competitive ratio in the algorithmic sense is based on the worst-case instance, while more modern literature sometimes uses the average case. In Definition 3.3, is the minimal number of queries for the worst-case instance?
>
> Yes, Definition 3.3 measures $Q_\mathcal{A}^f / Q_{\mathrm{opt}}^f$ per model $f$ and then takes the supremum over all $f$, i.e., the worst-case. Extending this to an average-case or learning-augmented framework, where one analyzes expected competitiveness under a distribution over models, is indeed a promising avenue, and will be highlighted as future work. However, note that very different methodological tools might be needed for such situations. In addition, note that our empirical study also provides average-case insights regarding the practical performances of our approach on real-world datasets.
>
> ---
> > What is the analysis of the competitive ratios of the existing baselines (e.g., Pathfinding, CF, Dual CF): are your empirical results showing something or just validating your theoretical results?
>
> Deriving formal competitive ratios for PathFinding, CF, or DualCF would require a detailed worst-case analysis for each method as well as a common target (a certain recovery threshold, or perfect fidelity). However, this is a non-systematic and difficult task. To compare these methods with our proposal, we rely on our empirical comparisons (e.g., Figures 3 to 5), which demonstrate that TRA consistently outperforms these baselines in query count and fidelity, thereby validating that TRA's competitive guarantees translate into practical superiority.
>
> ---
> > $m$, where most features are one-hot vs $m$ when features are continuous, seems like different levels of difficulty. Table 1 separates feature types-how does this impact your analysis? Proposition 3.6 seems loose if all $m$ features are one-hot.
>
> Indeed, the rightmost calculation in Proposition 3.6 corresponds to a "looser" (but more readable) worst-case estimate than the leftmost calculation, which explicitly models the possible number of split levels for each feature. This does not impact much the analysis since one can rely on the leftmost calculation for any specific case. Furthermore, Proposition 3.7 shows that this calculation is "tight" for any divide-and-conquer-based algorithm, by identifying a case in which this situation happens.
>
> ---
> > What happens if the oracle outputs noisy counterfactuals (wrong with some probability)?
>
> Thank you for raising this point. First, it is important to note that any CF can be verified with a single prediction query to ensure that $f(x')\neq f(x)$. If the counterfactual does not actually change the predicted class, it can be discarded and does not affect the algorithm’s operation. Note that in real-world deployments, unreliable counterfactual algorithms violate the "right to explanations" and should preferably not be used. Now, in cases where the counterfactual explanation is valid but suboptimal (i.e., it lies far from decision boundaries), TRA can still operate correctly. Specifically, such a counterfactual can be post-processed (through bisection line-searches over each feature using simple prediction queries) as mentioned in Appendix C into a locally-optimal explanation lying on the decision boundary. In summary, as long as the oracle is reliable (i.e., it returns a counterfactual whenever one exists), it provides a direction of change in prediction, which is sufficient to identify a nearby boundary. This mechanism enables TRA to function correctly even when the counterfactuals are noisy and sub-optimal.
>
> ---
>
> Once again, thank you for your positive appreciation and comments, we hope our responses have helped clarify any remaining gap. If so, we kindly invite you to reconsider your evaluation of the paper. We are also available for further clarification if any other doubts remain.

---

> ### Comment · Reviewer_hAR3 · 2025-08-04
>
> Dear paper authors,
> Thank you for your responses and clarification. They have sufficiently answered my questions. I appreciate it!

---

### Official Review · Reviewer_TuFm · 2025-06-30

**Clarity:** 3
**Significance:** 3
**Originality:** 3
**Rating:** 5
**Confidence:** 2

**Summary:**

The authors present an approach for model extraction attacks based on counterfactual explanations underlying the fact that explainability techniques may undermine the security of ML models.
The approach addresses models based on additive decision trees.

**Questions:**

I'd suggest authors to introduce earlier what they mean by functional equivalence guarantees and fidelity. They are formally defined later on in the article, but as they insist on these elements in the introduction, it would be nice to clarify the concept (to a wide audience as that of the current venue).

Does the system only considers binary trees construction ?

The authors discuss about the choice of the query points in the input space, but they do not discuss the importance of the local optimality of the counterfactual. How the system behaves in absence of such constraint.

Related work should be introduced prior to experimental setup as the authors already discuss representative work from STAR. The reader would also appreciate a discussion of the current STAR and how qualitatively (beyond quantitative results) the current proposal pushes the STAR one step further. CF and DualCF attacks employing counterfactuals were already considered by the community.

The way the algorithm behaves in absence of local or generic optimal counterfactual artifact should be included in the main article as the quality of the counterfactual is a point that rise questions in reader's mind. The results on page 16 in the annex seems that the non-optimal counterfactual explanations retured by DiCE helped fitting the surrogate models better. So if non-optimal works better, why not considering also random sampled counterfactuals ?

How the complexity of the surrogate models varies with regard to the nature of the counterfactuals used ? I understand that the main criteria here are  functional equivalence guarantees and fidelity, but mastering the complexity of the surrogate model might also be a concern. There are elements in supplementary material, but I suggest to push them in the main paper as the choice of the couternfactual is an element part of the process.

I suggest authors to reproduce the seeds used. In order to secure the conclusions, I would recommand to increase the number of runs conducted. Besides, variances for experiments were not reported.

**Ethical Concerns:**

["NO or VERY MINOR ethics concerns only"]

**Final Justification:**

The rebuttal carefully addressed all my concerns.

I m convinced now by the originality and the significance of the contribution. The clarity should also be improved considering promised updates to the final version.

**Quality:**

3

**Strengths And Weaknesses:**

The approach yields good results in comparaison with the current SOTA, exploiting features that are pushed forward by explainibility techniques without other assumption when considering models based on additive decision trees.
The evaluation seems sound without major concerned raised by the protocol adopted.

It is unclear how this can be generalized to other family of models where decision boundaries are not parralel to the axis.
It is important to clearly pinpoint the issue in additive decision trees spectrum, but it would be interesting to account for the amount of ML as a service providing this kind of models. Does this vulnerability impact niche platforms or mainstream ones ?

Elements from the Annex should have had their place in the main paper (such as B2 and C in relation to counterfactual optimality and D) in order to offer a complete view on the main contributions.

---

> ### Author Rebuttal · Authors · 2025-07-26
>
> We appreciate your thoughtful review!
>
> ---
> > It is unclear how [...] or mainstream ones ?
>
> Thank you for this interesting perspective! While most of the core ideas of our divide-and-conquer algorithm would still work in the case of models with non-axis-parallel decision boundaries recovered as oblique decision trees, a key challenge lies in the fact that the coordinates of a locally optimal counterfactual example no longer uniquely define a separating hyperplane. For instance, a counterfactual located at a corner of the decision boundary may correspond to multiple combinations of intersecting hyperplanes. In such cases, additional prediction queries would be needed to recover the decision boundaries, yet it remains unclear to us how to do this efficiently. This is, however, a very interesting research question that we will continue examining and mention explicitly in the perspectives of the paper. We are also not aware of any functionally equivalent approach targeting other families of models in the current literature.
>
> That said, focusing on axis-parallel models already uncovers a highly relevant vulnerability. Most tree-based algorithms (e.g., boosting ensembles or random forests as studied in our Result 3) construct axis-aligned splits by default and are widely used in production. Furthermore, many mainstream MLaaS platforms (e.g., Microsoft Azure, Google Cloud AI) offer decision tree-based models and support counterfactual explanations, making them directly susceptible to the attack we study. As such, the vulnerability we expose applies very broadly.
>
> ---
> > Elements from the Annex [...] on the main contributions.
>
> We completely agree. Due to the NeurIPS page limit, we had to provide some detailed analyses (specifically Sections B2, C, and D) in the appendix. These sections are explicitly referred to in the main text to ensure a complete presentation of our contributions. Nevertheless, the page limit is typically increased a bit for camera-ready paper preparation, so we could definitely migrate some of this content back to the main paper.
>
> ---
> > I'd suggest authors to introduce [...] the concept (to a wide audience as that of the current venue).
>
> Sure, we will include informal definitions of these notions earlier in the introduction as suggested.
>
> ---
> > Does the system only considers binary trees construction ?
>
> Formally, our proposed attack applies to any model with axis-parallel decision boundaries, which can always be represented by a decision tree, whether binary or not. While TRA encodes the extracted model as a binary decision tree, this is simply a representational choice: the resulting tree is guaranteed to be functionally equivalent to the target model, even if the latter uses a different tree structure (since multiple structurally different trees may provide the same prediction function).
>
> As mentioned in connection with one of your previous comments, it is a very interesting research perspective to extend our work to models with non-axis-parallel or piecewise-linear decision boundaries (e.g., ReLU-based neural networks).
>
> ---
> >The authors discuss about [...] of such constraint.
>
> As mentioned in Appendix C, any valid but possibly sub-optimal counterfactual explanation can be post-processed (through bisection line-searches over each feature using simple prediction queries) into a locally-optimal explanation lying on the decision boundary. Therefore, as long as the counterfactual oracle is reliable (i.e., it returns a counterfactual whenever one exists), the returned example reveals a direction in which the prediction changes, which is then sufficient to locate a nearby decision boundary. Consequently, the proposed attack framework remains effective in this setting, requiring only a few additional prediction queries. We will strengthen the discussion in the paper to clarify this aspect.
>
> ---
> >Related work should [...] by the community.
>
> In the current version of the paper, we chose to streamline the presentation to rapidly progress towards the method description, i.e., by locating the comprehensive discussion on STAR in Section 5, just before the conclusion. That said, contextualization of the work, discussions on the relative merit and capabilities of each method in the current STAR, have been also added in introduction and when presenting the baselines in the experimental section. We remain open to adjusting the structure of the paper to present related work earlier.
>
> Sure, we will also improve the discussions on both the quantitative and qualitative contributions of our approach. Empirically, TRA significantly outperforms existing counterfactual-based attacks such as CF and DualCF in terms of fidelity and query efficiency. From a theoretical and qualitative standpoint, TRA provides capabilities and formal guarantees achieved by no other approach:
> - Tramèr et al. [2016] do not leverage counterfactual explanations and make stronger assumptions about the model’s API. Their method is specific to decision-tree target models (e.g., excluding random forests) and often requires far more queries. In contrast, TRA uses orders of magnitude fewer queries by exploiting counterfactual explanations and can handle any model with axis-parallel decision boundaries.
> - Dissanayake and Dutta [2024] focus on target models with convex and Lipschitz-continuous decision boundaries, which excludes models with axis-parallel boundaries. Their approach relies on globally optimal counterfactuals (whereas TRA accommodates locally optimal ones) and, while it provides certified bounds on the decision boundary approximation error, it does not guarantee functional equivalence.
>
> Finally, note that neither CF nor DualCF provide any fidelity guarantees. We will strengthen the discussions in the paper to ensure these differences are very clearly stated.
>
> ---
> >The way the algorithm [...] sampled counterfactuals ?
>
> CF and DualCF build a surrogate dataset using counterfactual queries to the target model and then train a surrogate model on this dataset. In that context, DiCE counterfactuals provided greater diversity, which slightly improved surrogate performance. In contrast, TRA takes a fundamentally different approach. It uses counterfactual queries strategically, selecting them one by one within specific subregions of the input space to uncover decision boundaries with guaranteed fidelity and high query efficiency.
>
> While TRA benefits from locally optimal counterfactuals, it does not strictly require them: even sub-optimal counterfactuals can be post-processed to recover locally optimal points lying on decision boundaries. Appendix C details how to use TRA when the counterfactuals are not optimal and presents some results using a simple counterfactual heuristic.  These experiments show that TRA's performance remains robust regardless of the counterfactual generation method, as long as the oracle returns a valid counterfactual when one exists. We will definitely highlight this further in the main paper.
>
> ---
> > How the complexity [...] part of the process.
>
> You are absolutely right that, even once functional equivalence is achieved, the resulting surrogate model can vary in size depending on the order in which the decision boundaries are discovered. We will include the experimental results related to this metric in Appendix C, along with a discussion in the main paper. Importantly, we note that once a functionally equivalent model is obtained, one can apply standard model reformulation and compression techniques (e.g., ["Born-again tree ensembles", ICML, 2020]) to reduce the model size further. This step can be performed entirely offline, requiring no additional access to the prediction or counterfactual oracle, and would remain invisible to a MLaaS model provider.
>
> ---
> > I suggest authors to reproduce [...] were not reported.
>
> Thank you for this suggestion. Sure, the source code we provide along with the paper (including the seeds) will allow complete reproducibility of the results. Regarding the number of runs and variance reporting, it is important to note that our Tree Reconstruction Attack (TRA) is fully deterministic: given a fixed target model and counterfactual oracle, the algorithm produces identical outputs across runs. Consequently, the only source of variability in our experiments stems from the training of the target models (decision trees or random forests).
>
> In Figures 3 and 5, we plot **performance profiles** of fidelity vs.\ number of queries. At each query-count $q$, the curve shows
> $$\text{mean fidelity at }q = \frac1N \sum_{i=1}^N fidelity(\hat{f}_i,f_i),$$
> where $N$ is the number of independent target models (each with a different seed or max depth). By design, performance profiles display only the mean across all runs at each $q$; error bars had to be omitted for clarity of the graph. Furthermore, our approach quickly reaches perfect extraction fidelity 100\% across all runs, in that regime there is *no variability*, since fidelity is identically 1.0. We can complement these figures with additional tables in the Appendix, as illustrated here for TRA on the Adult dataset for target models of maximum depth 9:
>
>  | \# Queries|Mean fidelity $±$ Std Dev|
> |:-:|:-:|
> |$20$|$ 0.7053 ± 0.085$|
> |$60$|$0.8564 ± 0.044$|
> |$100$|$0.8942 ± 0.051$|
> |$500$|$0.9693 ± 0.010$|
> |$1000$|$0.9775 ± 0.010$|
> |$5000$|$0.9968 ± 0.003$|
> |$10000$|$0.9988 ± 0.001$|
> |$14920$|$1.0000 ± 0.000$|
>
> Finally, in Figure 4, each marker corresponds to a single independent extraction (i.e., a target model trained under one seed and configuration). Thus, variability is explicitly visualized via the scatter of individual data points rather than through aggregated statistics.
>
> ----
> Once again, thank you for your positive appreciation and comments, we hope our responses have helped clarify any remaining gap. If this is the case, please consider revising your evaluation of the paper. We are also available for further clarification if any other doubts remain.

---

> > ### Comment · Reviewer_TuFm · 2025-08-02
> >
> > I d like to thanks the authors for carefully addressing all my comments and remarks.
> >
> > I m satisfied by the complements brought.

---

### Official Review · Reviewer_349a · 2025-07-01

**Clarity:** 2
**Significance:** 3
**Originality:** 3
**Rating:** 4
**Confidence:** 4

**Summary:**

The paper proposes an online model extraction attack based on counterfactual explanations, specifically designed for decision tree and random forest models. The attack involves sequentially recovering the decision splits over each input feature using counterfactual explanations. A query complexity analysis for the algorithm (in terms of the tree depth and the input dimension) has also been presented. Theoretical guarantees for the performance of the algorithm has been given in terms of c-competitiveness w.r.t. an optimal offline algorithm.

**Questions:**

Instead of popping out the next available subspace from the list Q, will there be an improvement for the anytime performance if we pop the subspace with the largest volume?

Also, please see the concerns I have raised in the weaknesses above.

**Ethical Concerns:**

["NO or VERY MINOR ethics concerns only"]

**Final Justification:**

Authors have clarified the questions that I had, and also have agreed to revised the manuscript to resolve all the concerns I have raised. In light of this, I am happy to increase the rating.

**Limitations:**

Yes

**Paper Formatting Concerns:**

No major formatting concerns

**Quality:**

3

**Strengths And Weaknesses:**

**Strengths:**

1. The discussed vulnerability is very much related, specifically to the finance sector where decision trees and random forest models are used for reliability and transparency; which is also a context where counterfactual explanations play an important role. Therefore the problem discussed in the paper is highly relevant.
2. Provides an interesting algorithm with a theoretical analysis on the complexity and performance.
3. Experiments are comprehensive and intuitive. Experimental setups are clearly explained.
4. The paper content is well-organized.

**Weaknesses:**

*On the algorithm*
1. The algorithm relies strongly on the fact that the counterfactual explanations (CFs) provided by the oracle lie on the decision boundary (local/global optimality) which is usually a difficult factor to be satisfied (e.g.: robust counterfactual explanations – see [1]). Authors have mentioned this as a problem of DiCE in Appendix C (top of page 24), which, however should be thought of as a possible real-world scenario rather than an imperfection of CF generation.
2. Since the line joining the CF and the query instance (x’---x) may not be exactly parallel to many feature dimensions, the number of splits will be extremely large.
3. No clear definition of the splitting function. Does it exclude the boundaries of the original subspace and the split? Otherwise it seems like there can be situations where the algorithm will never reveal some splits.

*On the proofs*
1. Proposition 3.4: In the inductive step, authors mention that s_i^{(n_q)}=s_i^{(n+1)} for I \neq j and q \in {1,2}, which is not clear to me. Shouldn’t this be s_i^{(n_1)} + s_i^{(n_2)}=s_i^{(n+1)} ?
2. Proposition 3.6 upper bound: On page 13 top, the worst-case number of queries required is computed to be s_1+….. based on the fact that a checkerboard-like split would be the worst-case split possible. However, can we say that the recovery algorithm, i.e., first detecting the splits along feature 1 and then detecting the splits along feature 2 etc. is the worst-case algorithm?

*On the experiments*
1. In the first few experiments, OCEAN has been used as the CF generator for the proposed TRA attack, whereas DiCE has been used as the CF generator for the CF [Aivodji et al.] and DualCF [Wang et al.]. It is not clear how comparable the generated Cfs are.

*Other minor comments:*
1. The definition of “split levels” is not clear
2. In the proofs, re-iterating the statement (proposition or the corollary) might make it easier to read.
3. Including the KKT conditions in the proof of Corollary 3.5 would be good for the reader.


[1] Dutta, S., Long, J., Mishra, S., Tilli, C., & Magazzeni, D. (2022, June). Robust counterfactual explanations for tree-based ensembles. In International conference on machine learning (pp. 5742-5756). PMLR.

---

> ### Author Rebuttal · Authors · 2025-07-25
>
> We appreciate your thorough review. Please see our point‐by‐point responses below.
>
> ---
> > The algorithm relies strongly on the fact that the counterfactual explanations (CFs) provided by the oracle lie on the decision boundary (local/global optimality) which is usually a difficult factor to be satisfied (e.g.: robust counterfactual explanations – see [1]). Authors have mentioned this as a problem of DiCE in Appendix C (top of page 24), which, however should be thought of as a possible real-world scenario rather than an imperfection of CF generation.
>
> Indeed, this is an important point and application scenario, which can also be handled by TRA. As discussed in Appendix C, any valid counterfactual explanation can be post-processed (through bisection line-searches over each feature using simple prediction queries) into a locally-optimal explanation lying on the decision boundary. Consequently, as long as the counterfactual oracle is reliable (i.e., it returns a counterfactual whenever one exists), the returned example reveals a direction in which the prediction changes, which is sufficient to locate a nearby decision boundary. We will better clarify this important point and the line search process in the revised paper.
>
> Importantly, the key limitation of DiCE isn't that its counterfactuals lie off the boundary, but that it regularly fails to produce any counterfactual at all (even when one exists within the specified subspace).
>
> ---
> > No clear definition of the splitting function. Does it exclude the boundaries of the original subspace and the split? Otherwise it seems like there can be situations where the algorithm will never reveal some splits.
>
> > Since the line joining the CF and the query instance ($x’$---$x$) may not be exactly parallel to many feature dimensions, the number of splits will be extremely large.
>
> Thanks for pointing this out! Indeed, the paper provided limited details on the definition of the $split$ function, and we will improve this in the revision.
> The purpose of the $split$ function is to partition the current region based on the counterfactual example $x'$ returned by the oracle. If no counterfactual is found, the region is homogeneous (i.e., the prediction is constant), and a leaf is created, this is the base case. Otherwise, any locally-optimal counterfactual (possibly after post-processing, if needed) lies on a hyperplane (or vertex of a hyperrectangle) delimiting distinct prediction regions.
> Let $k$ denote the number of features on which $x'$ differs from the query point $x$. The $split$ function then partitions the region along one of these features, enqueues one of the resulting subregions, and continues splitting the remaining region (that contains $x'$) along the remaining $k-1$ features. This process results in exactly $k+1$ subregions. Consequently, by enabling a single counterfactual query to reveal multiple decision boundaries, the procedure ensures that no relevant split is missed and improves query efficiency. Regarding your second point, it is also important to note that this division stays linear in $k$; therefore, even if a counterfactual example requires changes along several features, the procedure remains efficient. We will definitely clarify this aspect in the paper and include a pseudo-code of the $split$ function in the appendix.
>
> ---
>
> > Proposition 3.4: In the inductive step, authors mention that $s_i^{(n_q)}=s_i^{(n+1)}$ for $i \neq j$ and $q \in {1,2}$, which is not clear to me. Shouldn’t this be $s_i^{(n_1)} + s_i^{(n_2)}=s_i^{(n+1)}$ ?
>
> As part of the inductive step, the feature space is divided along an arbitrary dimension $j$, and no new divisions are introduced in any other dimension $i\neq j$. Each child problem, therefore, has exactly the same number of splits in each dimension $i$ as the parent when $i\neq j$, so $s_i^{(n_q)} = s_i^{(n+1)} \quad (i\neq j,\,q\in\{1,2\})$.
>
> ---
>
> > Proposition 3.6 upper bound: On page 13 top, the worst-case number of queries required is computed to be $s_1+….$ based on the fact that a checkerboard-like split would be the worst-case split possible. However, can we say that the recovery algorithm, i.e., first detecting the splits along feature 1 and then detecting the splits along feature 2 etc. is the worst-case algorithm?
>
>
> The objective of the first part of the proof of Proposition 3.6 is to establish an upper-bound on the competitive ratio of TRA, defined as $C_{TRA}^{(n,m)} = sup_{f \in \mathcal{F}} (Q_{TRA}^f / Q_{opt}^f)$. To this end, we determine the smallest possible value for $Q_{opt}^f$, and the largest possible one for $Q_{TRA}^f$. Notably, the worst case for TRA occurs when the target model has a checkerboard-like structure where each axis-aligned cell in the feature space points to a distinct class. In such a configuration, recovering the model requires the maximum number of region splits and counterfactual queries, as every cell must be visited to ensure label consistency. Importantly, in this setting, the order in which TRA discovers the feature splits does not affect the total number of queries. This is because any axis-aligned split along one feature dimension necessarily intersects all splits along the remaining dimensions, so the total number of resulting subregions remains the same regardless of the order, and every cell in the checkerboard has to be tested individually for uniformity in the base case. Finally, Proposition 3.7 shows that this bound is tight: any divide-and-conquer extraction algorithm will face the same worst-case behavior.
>
> ---
>
> > In the first few experiments, OCEAN has been used as the CF generator for the proposed TRA attack, whereas DiCE has been used as the CF generator for the CF [Aivodji et al.] and DualCF [Wang et al.]. It is not clear how comparable the generated Cfs are.
>
> To ensure a complete and fair comparison with our approach, we evaluated the CF and DualCF baseline attacks using both DiCE and OCEAN as counterfactual generators in our preliminary experiments, even though the original papers used only DiCE. Then, we retained the best configuration for the baselines to put them in the most favorable light (in this case, using DiCE, as discussed in Section 4.1, lines 250–255). Additionally, Appendix B.2 provides a detailed comparison of their performance with both counterfactual oracles.
>
> ---
>
> > The definition of “split levels” is not clear.
>
> Thank you for pointing this out. For a given decision tree, the "split levels" of a feature refer to the set of threshold values used in split conditions involving that feature across the entire tree. For binary features, only a single split level is possible, while continuous features may involve multiple distinct threshold values. We will reformulate the definition (Section 3.2, lines 143-145) to clarify it.
>
> ---
>
> > In the proofs, re-iterating the statement (proposition or the corollary) might make it easier to read.
>
> Thank you for the suggestion. We agree that reiterating the statement can improve readability, and we will add a restatement of each proposition and corollary before its corresponding proof.
>
> ---
>
> > Including the KKT conditions in the proof of Corollary 3.5 would be good for the reader.
>
> Sure, we will definitely add them in the paper.
>
> ---
>
> > Instead of popping out the next available subspace from the list Q, will there be an improvement for the anytime performance if we pop the subspace with the largest volume?
>
> Indeed, while different orderings of the priority queue $\mathcal{Q}$ do not impact the total number of queries needed by the algorithm, they can yield different anytime fidelity performances. This is precisely why we mentioned the design of "alternative priority strategies" as an insightful direction (Section 3.1, line 201-204). In particular, prioritizing subspaces by largest volume is one possibility, and is indeed likely to improve TRA's anytime performance.
>
> ---
>
> Once again, thank you for your comments, we hope our responses have helped clarify any remaining gap. If so, we kindly invite you to reconsider your evaluation of the paper. We remain available to provide any further clarification should additional questions arise.

---

> > ### Comment · Reviewer_349a · 2025-08-06
> >
> > I appreciate the detailed response by the authors which clarifies and addresses all the questions and concerns I had. Consequently, in light of the proposed revisions to the manuscript, I have updated the rating.

---

### Official Review · Reviewer_oRDk · 2025-07-02

**Clarity:** 3
**Significance:** 3
**Originality:** 4
**Rating:** 5
**Confidence:** 3

**Summary:**

This paper proposes a novel model extraction attack for additive decision tree-based models. This attack provably achieves perfect fidelity while relying only on the availability of a locally optimal counterfactual explanation oracle, a commonly used explainability technique. The paper further analyzes the worst-case query complexity of the attack and develops a theoretical framework based on competitive analysis to evaluate the efficiency of model extraction attacks. This framework shows that the proposed algorithm is optimal among the class of divide and conquer algorithms. Experimental results evaluate the anytime performance of the attack and show that the algorithm is superior to existing tree-based model extraction attacks.

**Questions:**

-	What is the reason that $m$ is not necessarily equal to $m_N + m_B + m_C + m_D$ in Table 1?
-	How does the competitive ratio of Algorithm 1 compare to non divide and conquer based methods?
-	Have you thought about possible defense methods against the attack from Algorithm 1?

**Ethical Concerns:**

["NO or VERY MINOR ethics concerns only"]

**Final Justification:**

The authors clearly answered the questions I and other reviewers had. Therefore, I am keeping the rating to be the same i.e. an Accept since I believe the paper meets the quality standard of NeurIPS.

**Limitations:**

yes

**Quality:**

3

**Strengths And Weaknesses:**

$\large \textbf{Quality}:$

This paper is of a good quality and comprehensive in both its experimental evaluation and theoretical analysis. The experimental evaluation includes standard binary classification datasets from literature for tree-based models and provides results which demonstrate the perfect fidelity and compares the anytime performance. Theoretically, the idea to use competitive analysis is beneficial and simplifies the analysis while answering a fundamental question in model extraction attacks on the number of queries for perfect fidelity extraction.

It would be beneficial to also include some results on the competitive ratio to show experimentally the results from Propositions 3.6 and 3.7. Similarly, understanding if the results can be modified to account for some target level of fidelity as opposed to perfect fidelity would be helpful and correspond to more realistic model extraction attacks such as in the case of large models or models beyond trees.

$\large \textbf{Clarity}:$

The paper is generally clear and easy to understand. The figures and numerical examples help clarify the theoretical results and the proofs are written in a way that they can be easily followed. The experimental results are organized nicely with the main conclusions listed and the corresponding figures referenced.

Providing some more intuition for the definition of a counterfactual would be beneficial. As someone not too familiar with decision trees, I was able to understand from the numerical example and Figure 1 but clarifying or providing some more discussion in Definition 3.1 would be beneficial.

$\large \textbf{Significance and Originality}:$

The results from the paper are helpful in developing a rigorous understanding for model extraction attacks against tree-based models which provide counterfactual explanations. The ideas used are original and it is interesting to see the concept of competitive analysis and online optimization be applied to model extraction. The algorithm, while based on a simple idea, is interesting and the properties of optimality and perfect fidelity make it superior to existing model extraction algorithms for decision trees. The theoretical results are helpful to benchmark the performance of model extraction algorithms and understand their limitations. While the scope of the paper is limited to decision tree-based models, these ideas and techniques may be carried over to standard machine learning models. Moreover, the motivation around counterfactual explanations is practically relevant as model explainability methods gain importance.

---

> ### Author Rebuttal · Authors · 2025-07-25
>
> Thank you for the insightful feedback! Below are our point‐by‐point responses (with planned edits highlighted).
>
> ---
>
> > It would be beneficial to also include some results on the competitive ratio to show experimentally the results from Propositions 3.6 and 3.7.
>
> Sure, as seen in our experiments, the number of counterfactual queries empirically required by TRA to achieve functionally equivalent model extraction is significantly smaller than its theoretical worst-case obtained through Proposition 3.6. For instance, the following table compares these values for each dataset and for decision trees of maximum depth 9 (average, over $5$ random seeds). We will include these additional comparisons in the Appendix with related comments in Section 4.2.
>
> | Dataset       | Empirical \# queries   | Worst‐case \# queries        |
> | ------------- | :-----------------------------------------: | :--------------------------------------------: |
> | SPerformance | $1.16e+03$ | $3.02e+08$ |
> | Adult | $3.70e+04$ | $2.45e+14$ |
> | German Credit | $5.18e+01$ | $2.86e+03$ |
> | Credit Card | $6.97e+04$ | $6.49e+11$ |
> | COMPAS | $1.53e+02$ | $1.07e+03$ |
>
>
> ---
>
> > Understanding if the results can be modified to account for some target level of fidelity as opposed to perfect fidelity would be helpful [...].
>
> Indeed, as discussed in Section 3.2, TRA is an anytime algorithm, i.e., it can return a surrogate model at any point during its execution. Figures 3 and 5 show how surrogate's fidelity improves over the iterations. Note that fidelity must be measured for a certain sample distribution. In the case of a uniform distribution over the feature space, we can lower-bound the fidelity by the fraction of total volume corresponding to the leaf regions that have already been fully explored by TRA at a given iteration, and stop whenever a given target is met. We will mention this possibility in the revised paper.
>
> ---
>
> > Providing some more intuition for the definition of a counterfactual would be beneficial
>
> Sure, in the revised paper, we will better discuss the concept of counterfactual explanations at a high level right before Definition 3.1.
>
> ---
>
> > What is the reason that $m$ is not necessarily equal to $m_N + m_B + m_C + m_D$ in Table 1?
>
> In Table 1 each of the $m_C$ categorical features is one-hot encoded into $c_j$ binary dimensions, where $c_j$ is the number of categories of feature $j$. As a result, the total number of features becomes:
> $$
>   m = m_N + m_B + m_D + \sum_{j=1}^{m_C} c_j.
> $$
>
> We will clarify this point in the paper by adding a footnote to Table 1.
>
> ---
>
> > How does the competitive ratio of Algorithm 1 compare to non divide and conquer based methods?
>
> Thanks, this is an interesting but very challenging question. Analyzing the competitive ratio of any given algorithm is a non-systematic and difficult task, as it requires bounding the worst-case performance gap between an online algorithm and an omniscient offline counterpart over all possible inputs. This is also why proofs of competitive ratio are valuable from a theoretical standpoint.
> To the best of our knowledge, there are no known theoretical results regarding the competitive ratio of other model extraction attacks. Our work provides the first such theoretical result in this context.
>
> ---
>
> > Have you thought about possible defense methods against the attack from Algorithm 1?
>
> Thank you for raising this important point. As long as the counterfactual oracle reflects the true model, returns a valid explanation whenever one exists (a property typically required in practical applications), and the number of queries is not constrained, TRA will ultimately converge to a functionally equivalent model extraction. Nonetheless, service providers might be able to mitigate the attack by rate-limiting the number of queries per user, introducing human verification methods during queries (e.g., CAPTCHAs), or monitoring for algorithmically generated query patterns. When such patterns are detected, they could throttle responses or alter the oracle behavior to reduce the attack's effectiveness. We will include a brief discussion of potential countermeasures in the appendix.
>
> ---
>
> Overall, thank you for your comments and your appreciation of the paper, we hope our responses have helped clarify any remaining gap. We remain available to provide any further clarification should additional questions arise.

---

> > ### Comment · Reviewer_oRDk · 2025-08-04
> >
> > Thank you for the detailed responses to my questions. Regarding the first answer, you mention that the number of queries required is significantly smaller than the theoretical worst case but the table below does not seem to show the same result. For the last 4 of the 5 datasets, the empiricial number of queries is larger than the worst case value.

---

> > > ### Author Response · Authors · 2025-08-04
> > >
> > > Thank you for your feedback.
> > >
> > > Regarding the comparison between the empirical number of queries required by TRA to achieve functionally equivalent model extraction and its theoretical worst-case, we would like to clarify that the numbers reported in the table are written in scientific notation, where "$e+XX$" means "$\times 10^{XX}$". As such, the empirical number of queries is consistently smaller than the corresponding theoretical worst-case. For clarity, we reproduce the same table below with the numbers written in standard (full) notation:
> > >
> > > | Dataset | Empirical \# queries | Worst-case \# queries |
> > > |:-------:|:-------:|:-------:|
> > > | SPerformance | $1.16 \times 10^3$ | $3.02 \times  10^8$ |
> > > | Adult | $3.70 \times 10^4$ | $2.45 \times 10^{14}$ |
> > > | German Credit | $5.18 \times 10^1$ | $2.86 \times 10^3$ |
> > > | Credit Card | $6.97 \times 10^4$ | $6.49 \times 10^{11}$ |
> > > | COMPAS | $1.53 \times 10^2$ | $1.07 \times 10^3$ |

---

> > > > ### Comment · Reviewer_oRDk · 2025-08-04
> > > >
> > > > Thank you, my bad I had not looked at the numbers carefully.

---

### Decision · Program_Chairs · 2025-09-17

**Decision:**

Accept (spotlight)

**Comment:**

This paper proposes a new framework for model extraction attacks against tree-based models, introducing Tree Reconstruction Attack, a divide-and-conquer method using counterfactual oracles. The paper contributes both a rigorous theoretical foundation via competitive analysis and an efficient practical algorithm achieving provable fidelity guarantees.

Strengths include the combination of novel theory and practical relevance, strong experiments across datasets, and clear articulation of vulnerabilities in MLaaS platforms. The unanimous positive ratings after discussion, combined with the authors' thorough responses and commitment to improvements, support acceptance.

The reviewers and AC encourage the authors to include their responses of the rebuttal in the camera-ready version, not limited to:
1. Move key content from the appendix into the main text
2. Clarify definitions of functional equivalence guarantees and fidelity